# Towards Foundational Models for Molecular Learning on Large-Scale Multi-Task Datasets

**Dominique Beaini**[1,2,3]   **Shenyang Huang**[1,2,4]   **Joao Alex Cunha**[5]   **Zhiyi Li**[5]
**Gabriela Moisescu-Pareja**[1,2,4]   **Oleksandr Dymov**[1,2,3]   **Samuel Maddrell-Mander**[5]
**Callum McLean**[5]   **Frederik Wenkel**[1,2,3]   **Luis Müller**[7]   **Jama Hussein Mohamud**[1]
**Ali Parviz**[1,2,6]   **Michael Craig**[2]   **Michał Koziarski**[1,3]   **Jiarui Lu**[1,3]
**Zhaocheng Zhu**[1,3]   **Cristian Gabellini**[2]   **Kerstin Klaser**[5]   **Josef Dean**[5]
**Cas Wognum**[2]   **Maciej Sypetkowski**[2]   **Guillaume Rabusseau**[1,3,9]
**Reihaneh Rabbany**[1,4,9]   **Jian Tang**[1,8,9]   **Christopher Morris**[7]   **Mirco Ravanelli**[1,3]
**Guy Wolf**[1,3,9]   **Prudencio Tossou**[2]   **Hadrien Mary**[2]   **Therence Bois**[2]
**Andrew Fitzgibbon**[5]   **Błażej Banaszewski**[5]   **Chad Martin**[5]   **Dominic Masters**[5]

[1]Mila - Québec AI Institute    [2]Valence Labs    [3]Université de Montréal
[4]McGill University    [5]Graphcore    [6]New Jersey Institute of Technology
[7]RWTH Aachen University    [8]HEC Montréal    [9]CIFAR AI Chair

## Abstract

Recently, pre-trained foundation models have enabled significant advancements in multiple fields. In molecular machine learning, however, where datasets are often hand-curated, and hence typically small, the lack of datasets with labeled features, and codebases to manage those datasets, has hindered the development of foundation models. In this work, we present seven novel datasets categorized by size into three distinct categories: ToyMix, LargeMix and UltraLarge. These datasets push the boundaries in both the scale and the diversity of supervised labels for molecular learning. They cover nearly 100 million molecules and over 3000 sparsely defined tasks, totaling more than 13 billion individual labels of both quantum and biological nature. In comparison, our datasets contain 300 times more data points than the widely used OGB-LSC PCQM4Mv2 dataset, and 13 times more than the quantum-only QM1B dataset. In addition, to support the development of foundational models based on our proposed datasets, we present the Graphium graph machine learning library which simplifies the process of building and training molecular machine learning models for multi-task and multi-level molecular datasets. Finally, we present a range of baseline results as a starting point of multi-task and multi-level training on these datasets. Empirically, we observe that performance on low-resource biological datasets shows improvement by also training on large amounts of quantum data. This indicates that there may be potential in multi-task and multi-level training of a foundation model and fine-tuning it to resource-constrained downstream tasks. The Graphium library is publicly available on Github and the dataset links are available in Part 1 and Part 2.

## 1 Introduction

Graph and geometric deep learning models have been a key component in the recent successes of machine learning in drug discovery (Gasteiger et al.; Masters et al., 2023; Rampášek et al., 2022; Ying et al., 2021). These methods have demonstrated considerable performance in molecular representation learning (in the 2D (Rampášek et al., 2022), 3D (Gasteiger et al.; 2021), and 4D (Wu et al., 2022) cases), activity and property prediction (Huang et al., 2021), force field development (Batatia et al., 2022), molecular generation (Bilodeau et al., 2022), and the modeling of atomistic interactions (Corso et al., 2022). Like other deep learning methods, they require significant training data for high modeling accuracy. However, in the current therapeutics literature, most training datasets have limited samples (Huang et al., 2021). Tantalizingly, recent progress in self-supervised learning and

foundation models in natural language processing (NLP) (Brown et al., 2020a; Liu et al., 2023b) and computer vision (CV) (Dosovitskiy et al.) has drastically improved data efficiency in deep learning. In fact, by investing upfront in pre-training large models with lots of data, a one-time cost, it is proven that the learned inductive bias lowers the data requirements for downstream tasks.

Following these successes, many studies have explored the pre-training of large molecular graph neural networks and their benefits in low-data molecular modeling (Lin et al., 2022; Méndez-Lucio et al., 2022; Zhou et al., 2023). However, due to the scarcity of large and labeled molecular datasets, these studies could only leverage self-supervised techniques such as contrastive learning, auto-encoders, or denoising tasks (Hu et al., 2020a; Xia et al., 2023). Low-data modeling efforts by fine-tuning from these models have, as yet, yielded just a fraction of the advancement achieved by self-supervised models in NLP and CV (Sun et al., 2022). This is partially explained by the underspecification of molecules, and their conformers, as graphs, as their behavior is environment-dependent and mainly governed by quantum physics. For example, it is well known that structurally similar molecules may have vastly different bioactivity, known as *activity cliffs*, and this limits graph modeling based only on structural information (van Tilborg et al., 2022). We argue that building effective foundation models for molecular modeling requires supervised training with *both* quantum mechanical (QM) descriptions *and* biological environment-dependent data.

## 1.1 CONTRIBUTIONS

This study advances molecular research in three ways. First, we introduce a new family of multi-task datasets, orders of magnitude larger than the current state-of-the-art. Second, we describe a graph machine learning library called `Graphium` to facilitate efficient training on these extensive datasets. Third, a range of baseline models are implemented, supporting the value of training on a diverse collection of tasks.

**Datasets** We introduce three extensive and meticulously curated multi-label datasets that cover nearly 100 million molecules and over 3000 sparsely defined tasks, totaling more than 13 billion individual labels, currently the largest of their kind. These datasets have been designed for the supervised training of foundation models by combining labels representing quantum and biological properties acquired through both simulation and wet lab experimentation. The labels are also multi-level, encompassing both node-level and graph-level tasks. The diversity of labels facilitates efficient transfer learning and enables the construction of foundational models by improving their generalization ability for a wide range of downstream molecular modeling tasks.

To create these comprehensive datasets, we carefully curated and enhanced existing data with additional information. As a result, each molecule in our collection is accompanied by descriptions of its quantum mechanical (QM) properties and/or biological activities. The QM properties cover energetic, electronic, and geometric aspects, computed using various advanced methods, including density functional theory (DFT) methods like B3LYP (Nakata & Shimazaki, 2017), as well as semi-empirical methods like PM6 (Nakata et al., 2020). On the biological activity side, our datasets include molecular signatures obtained from dose-response bioassays, gene expression profiling, and toxicological profiling, as depicted in Figure 1. The joint modeling of quantum and biological effects promotes the ability to describe complex environment-dependent properties of molecules that would be infeasible to extract from what are typically limited experimental datasets.

**The `Graphium` Library** We have developed a comprehensive graph machine learning library called `Graphium` to facilitate efficient training on these extensive multi-task datasets. This novel library simplifies the process of building and training molecular graph foundation models by incorporating feature ensembles and complex feature interactions. By treating features (both positional and structural) and representations as fundamental building blocks and implementing state-of-the-art GNN layers, `Graphium` overcomes the challenges posed by existing frameworks that were primarily designed for sequential samples with limited interactions among node, edge, and graph features. Additionally, by providing features such as dataset combination, missing data handling, and joint training, `Graphium` handles the critical and otherwise complex engineering of training models on large dataset ensembles in a straightforward and highly customizable way.

**Baseline Results** We train a range of models in both single-dataset and multi-dataset scenarios for the dataset mixes presented. These provide solid baselines that can help guide future users of these datasets, but also give some indication of the value of training in this multi-dataset paradigm.

Figure 1: Visual summary of the proposed collections of molecular datasets. The "mixes" are meant to be predicted simultaneously in a multi-task fashion. They include quantum, chemical, and biological properties, categorical and continuous data points, graph-level and node-level tasks.

Specifically, the results on these models show that training of low-resource tasks can be significantly improved by training in combination with larger datasets.

In summary, our study introduces the largest 2D molecular datasets to date, specifically designed to train foundation models that can effectively comprehend molecules' quantum properties and biological adaptability and therefore be fine-tuned to a wide range of downstream tasks. In addition, we have developed the `Graphium` library to streamline the training process of these models and show a range of baseline results that highlight the effectiveness of both the datasets and library presented.

## 1.2 RELATED WORK

**Molecular Datasets** Several molecular datasets are available for node- and graph-level supervised learning tasks. A few notable examples are MoleculeNet (Wu et al., 2018), which is a collection of smaller datasets; QM9, which consists of quantum properties of 134k molecular conformers (Ramakrishnan et al., 2014; Wu et al., 2018); ZINC comprising the difference between the partition coefficient (clogP) and the synthetic accessibility score (SAS) of 12k molecules (Dwivedi et al., 2020); PCQM4M(v2) including the HOMO-LUMO gap (the lowest excitation energy of a molecule) of $\sim 3.6$M graphs (Hu et al., 2021b). Furthermore, Open Graph Benchmark's (OGB) (Hu et al., 2020b) large molecular datasets are OGBG-MOLHIV ($\sim$40k) and OGBG-MOLPCBA ($\sim$437k). And PCQM4M, the largest dataset, is still inadequate for large-scale supervised molecular pre-training, especially since it only contains a single property per molecule.

Several examples of large-scale datasets for self-supervised training exist, notably the 1.2 million molecule GUACAMOL(Brown et al., 2019) dataset, and the $\sim 880$ million molecule ZINC20 dataset (Irwin et al., 2020), and while these provide a valuable training platform they lack experimental labels. Examples of curated self-supervised datasets of 72 million molecules are used in Winter et al. (2019), however these datasets typically use computable quantum or chemical properties rather than environment-dependent labels. Some work in this direction, as proposed by Mayr et al. (2018), shows supervised datasets can be extracted from the large CHEMBL dataset, and in Méndez-Lucio et al. (2022) a dataset of some 0.5 million molecules with over 1000 labels is produced. Recent work on accelerating DFT calculations in Mathiasen et al. (2023) offers a tantalizing avenue to increasing the size of available 3D quantum datasets with 1 billion data points, albeit quantum-only. Finally, the GDB17 dataset (Ruddigkeit et al. (2012)) does enumerate some 166 billion chemically possible molecules, but does not provide labels or features associated with them.

However, none of these datasets are both sufficiently large, and provided with enough high quality experimental labels, to reflect the indirect properties relevant to developing a foundation model for drug discovery.

As such, our dataset collection departs from previous ones as it encompasses multiple graph-level and node-level tasks, in addition to complex environment-dependent properties.

**Pre-trained Chemical Foundation Models** Recent works explored pre-training for molecular property prediction via self-supervised learning techniques such as denoising (Zaidi et al., 2022); auto-encoders (Honda et al., 2019) and contrastive learning (Suresh et al., 2021). For a comprehensive

overview of self-supervised approaches to pre-training on molecular graphs, see Xia et al. (2023). As already discussed in Section 1, self-supervised pre-training does not take into account the quantum and biochemical characteristics of the molecules, necessitating pre-training in a supervised fashion. There have been some efforts at few-shot learning (Stanley et al., 2021; Schimunek et al., 2023), but the applicability of such methods is still limited both by a proper molecular featurization and a proper task featurization, the former of which we aim to tackle in our current work.

**Graph Learning Libraries** In recent years, many libraries have emerged to speed up the development of graph-learning models. PyTorch Geometric (PyG) (Fey & Lenssen, 2019) is built on PyTorch and provides essential functions for graph machine learning, including a custom data format for fast processing and loading of graph data, as well as various model implementations. DGL (Wang et al., 2019) is a general-purpose graph learning library that works with PyTorch, TensorFlow, or Jax and supports various model implementations and a custom data loading format. Jraph (Godwin* et al., 2020) for Jax, tf_geometric (Hu et al., 2021a) for TensorFlow, StellarGraph (Data61, 2018) built on Keras, GraphNets (Battaglia et al., 2018) for TensorFlow and Sonnet, and CogDL (Cen et al., 2023) are other libraries that provide building blocks for graph learning operations. In addition, GraphGym (You et al., 2020) is an experimental platform for graph learning experiments and model design, based on PyG. Although these libraries provide the fundamental components for creating and training graph learning models, extra engineering work is essential to construct scalable training and inference pipelines. In addition, there are specialized libraries that cater to drug discovery, including TorchDrug (Zhu et al., 2022) and DeepChem (Ramsundar et al., 2019). At the same time, GraphGPS (Rampášek et al., 2022) is dedicated to experiment and model design for graph transformers, and DIG (Liu et al., 2021) offers a toolkit for experimental design and model evaluation.

However, training and inference across multiple tasks and levels are yet to be explored in the context of molecular learning. Furthermore, existing libraries fail to tackle the difficulties of pre-training large-scale models, such as loading and managing large amounts of graph data efficiently. Thus, our datasets are accompanied by the `Graphium` library to facilitate the use of state-of-the-art GNNs in a large-scale multi-task setting. `Graphium` enables researchers to effortlessly apply GNN models to large-scale molecular datasets while paving the way towards foundational GNNs for chemistry.

## 2 PROPOSED DATASETS

Instead of proposing a single task dataset, we suggest a collection or "mix" of datasets enabling multi-task learning across relevant chemical tasks helpful in learning quantum chemistry and biochemistry and building foundation models. The datasets are visualised in Figure 1, complemented with statistics in Table 1. See Appendix B.2 and the Reproducibility section for details on datasets' licenses and availability, and see Appendix D for a deeper dive into the datasets.

Firstly, we propose a smaller TOYMIX dataset to enable fast iterations of the architecture while also providing valuable insight. Secondly, we propose the LARGEMIX, a mix of well-curated datasets with data for millions of chemical compounds over thousands of biological or quantum labels. Finally, we propose the ultra-large PM6_83M dataset, which pushes the limit of dataset scale with over 12 billion labels defined for 83 million molecules. In the following section, the three dataset categories are described in detail.

Table 1: Dataset statistics. G. / N. denote graph-level / node-level tasks, respectively, and C., R., and RC. denote *classification*, *regression* and *ranked classification* respectively. See Table 5 for detailed statistics.

|  | Dataset | Type | Task | # mol. | # G. labels | # G. data points | # N. labels | # N. data points |
|---|---|---|---|---|---|---|---|---|
| TOYMIX | QM9 | Quantum | R. | 134k | 19 | 2.5M | - | - |
|  | ZINC12K | Computed | R. | 12k | 3 | 36k | - | - |
|  | TOX21 | Bio-assays | C. | 8k | 12 | 75k | - | - |
| LARGEMIX | PCQM4M_G25_N4 | Quantum | R. | 3.8M | 25 | 93M | 4 | 197.7M |
|  | PCBA_1328 | Bio-assays | C. | 1.6M | 1,328 | 224.4M | - | - |
|  | L1000_VCAP | Transcriptomics | RC. | 15k | 978 | 15M | - | - |
|  | L1000_MCF7 | Transcriptomics | RC. | 12k | 978 | 11M | - | - |
| ULTRALARGE | PM6_83M | Quantum | R. | 83M | 62 | 4,003M | 7 | 8,509.2M |

## 2.1 TOYMIX (QM9 + TOX21 + ZINC12K)

The TOYMIX dataset combines the QM9, TOX21, and ZINC12K datasets. These datasets are well-known in the literature and used as toy datasets, or very simple datasets, in various contexts to make it easy to iterate quickly on models. By regrouping toy datasets from quantum ML, drug discovery, and GNN expressivity, we hope that the learned model will be representative of the model performance we can expect on the larger datasets. As can be seen in Figure 1, the TOYMIX collection contains 34 labels per molecule across the three datasets, containing a mix of floating point properties and class labels.

**Train/Validation/Test Splits** for all the datasets in TOYMIX are split randomly with a ratio of 0.8/0.1/0.1. Random splitting, rather than the more complex schemes we discuss below, is used since it is the simplest and fits the idea of having a toy dataset well.

**QM9** is a well-known dataset in the field of 3D GNNs (Gasteiger et al., 2021; Gasteiger et al.). It consists of 19 graph-level quantum properties associated to an energy-minimized 3D conformation of the molecules (Ramakrishnan et al., 2014). It is considered a simple dataset since all the molecules have at most 9 heavy atoms. We chose QM9 in our TOYMIX since it is very similar to the larger proposed quantum datasets, PCQM4M_multitask and PM6_83M, but with smaller molecules.

**Tox21** is a well-known dataset for researchers in machine learning for drug discovery (Huang et al., 2021; Tice et al., 2013). It consists of a multi-label classification task with 12 labels, with most labels missing and a strong imbalance towards the negative class. We chose TOX21 in our TOYMIX since it is very similar to the larger proposed bioassay dataset, PCBA_1328_1564K both in terms of sparsity and imbalance and to the L1000 datasets in terms of imbalance.

**ZINC12k** is a commonly used dataset in the GNN expressivity literature, especially for evaluating positional encodings, graph Transformers, and message passing for complexes (Beaini et al., 2021; Dwivedi et al., 2020; Rampášek et al., 2022; Bodnar et al., 2021). We include it in our TOYMIX as substructure understanding and positional encodings are important for generalization, especially in the case of graph Transformers where structural biases are missing. Hence, we hope that performance on this task will correlate well with performance when scaling graph Transformers.

## 2.2 LARGEMIX (PCQM4M + PCBA1328 + L1000)

In this section, we present the LARGEMIX dataset, comprised of four different datasets with tasks taken from quantum chemistry (PCQM4M), bio-assays (PCBA) and transcriptomics.

**Train/validation/test/test_seen Splits** For the PCQM4M_G25_N4, we create a 0.92/0.04/0.04 split. Then, for all the other datasets in LARGEMIX, we first create a "test_seen" split by taking the set of molecules from L1000 and PCBA1328 that are also present in the training set of PCQM4M_G25_N4, such that we can evaluate whether having the quantum properties of a molecule helps generalize for biological properties. For the remaining parts, we split randomly with a ratio of 0.92/0.04/0.04.

**L1000 VCAP and MCF7** The LINCS L1000 is a database generated using high-throughput transcriptomics. More than 30,000 perturbations on a set of 978 landmark genes (Subramanian et al., 2017) from multiple cell lines were screened. VCAP and MCF7 are, respectively, prostate cancer and human breast cancer cell lines. In L1000, most of the perturbagens are chemical, meaning that small drug-like molecules are added to the cell lines to observe how the gene expressions change. This allows the generation of biological signatures of the molecules, which are known to correlate with drug activity and side effects (Wang et al., 2016). To process the data into our two datasets comprising the VCAP and MCF7 cell lines, we used their "level 5" data composed of the cleanup data converted to z-scores (Subramanian et al., 2017), and applied a filter to keep only chemical perturbagens. However, we were left with multiple data points per molecule since some variables could change (e.g., incubation time) and generate a new measure. Given our objective of generating a single signature per molecule, we decided to take the measurements with the strongest global activity such that the variance over the 978 genes is maximal. Then, since these signatures are generally noisy, we binned them into five classes corresponding to z-scores based on the thresholds $\pm 2$.

The cell lines VCAP and MCF7 were selected since they have a higher number of unique molecule perturbagens than other cell lines. They also have a relatively low data imbalance, with $\sim$92% falling in the "neutral class" (where the z-score is between -2 and 2).

**PCBA1328** This dataset is very similar to the OGBG-PCBA dataset (Hu et al., 2020b), but instead of being limited to 128 assays and 437k molecules, it comprises 1,328 assays and 1.56M molecules. This dataset is very interesting for pre-training molecular models since it contains information about a molecule's behavior in various settings relevant to biochemists, with evidence that it improves binding predictions (Laufkötter et al., 2019). Analogous to the gene expression, we obtain a bio-assay-expression of each molecule. To gather the data, we have looped over the PubChem index of bioassays (Kim et al., 2016) and collected every dataset such that it contains more than 6,000 molecules annotated with either "Active" or "Inactive" and at least 10 of each. Then, we converted all the molecular IDs to canonical SMILES and used that to merge all of the bioassays into one dataset.

**PCQM4M_G25_N4** This dataset comes from the same data source as the OGBG-PCQM4M dataset, famously known for being part of the OGB large-scale challenge (Hu et al., 2021b) and being one of the only graph datasets where pure Transformers have proven successful (Luo et al., 2022; Ying et al., 2021). The data source is the PubChemQC project (Nakata & Shimazaki, 2017) that computed DFT properties on the energy-minimized conformation of 3.8M small molecules from PubChem. In contrast to the OGB challenge, we aim to provide enough data for pre-training graph ML models, so we do not limit ourselves to the HOMO-LUMO gap prediction (Hu et al., 2021b). Instead, we gather properties directly given by the DFT calculations (e.g., energies) and compute other 3D descriptors from the conformation (e.g., inertia, the plane of best fit). We also gather node-level properties, in this case the Mulliken and Lowdin charges at each atom. Furthermore, about half of the molecules have time-dependent DFT calculations to help inform about the molecule's excited state. Looking forward, we plan on adding edge-level tasks to enable the prediction of bond properties, such as their lengths and the gradient of the charges.

## 2.3 ULTRALARGE DATASET

**PM6_83M** This dataset is similar to PCQM4M and comes from the same PubChemQC project. However, it uses the PM6 semi-empirical computation of the quantum properties, which is orders of magnitude faster than DFT computation at the expense of lower accuracy (Nakata & Shimazaki, 2017; Nakata et al., 2020).

This dataset covers 83M unique molecules, 62 graph-level tasks, and 7 node-level tasks. To our knowledge, this is the largest dataset available for training 2D-GNNs regarding the number of unique molecules. The various tasks come from four different molecular states, namely "S0" for the ground state, "T0" for the lowest energy triplet excited state, "cation" for the positively charged state, and "anion" for the negatively charged state. There are 221M PM6 computations (Nakata et al., 2020).

## 3 THE GRAPHIUM LIBRARY

Drug discovery is a field with rich, multi-faceted datasets. Data is often inherently multi-tasked, containing label information at various levels, from nodes and edges to pairs of nodes and entire graphs. Hence, training models on such complex datasets requires a strategy for jointly learning on multiple task levels, presenting a unique challenge. To leverage the rich information contained in these dataset mixes and to benefit from their scale, dedicated software is necessary to facilitate efficient training in a multi-task fashion. To this end, we introduce the `Graphium` library, designed explicitly for large-scale multi-task and multi-level ML models within the molecular domain.

Key features of the `Graphium` library are summarised here, with further details to be found in Appendix E, with the relevant sections indicated in the text. The entire training procedure is defined in a modular configuration file, for example this allows users to swap out architecture, datasets or metrics for quick experimentation. The core novel feature of the `Graphium` library is the multi-level multi-task learning that facilitates training a model over multiple datasets with disparate and sparse labels through modular dataloading, task head, and loss functions. More details can be found in Appendix E.1. The flexible and modular modeling is detailed in Appendix E.2. Positional encoding methods, which provide node level information about location within the subgraph, are integral to many advanced molecular models. Key methods, including random walk and laplacian eigenvectors,

are provided in `Graphium`, this is detailed in Appendix E.3. Label normalisation (Appendix E.4), ranked classification loss (Appendix E.5) and the handling of missing data (Appendix E.6) are important details of the library that facilitate the combination of multiple sources of data across a range of tasks and the inevitable sparsity in the dataset. To make training large models resource efficient much attention is paid to the possibility of tuning model parameters on small models and applying these to larger models. One such approach called $\mu$P introduced in Yang et al. (2022) is included in `Graphium` to reduce the cost of hyper-parameter tuning. More details are given in Appendix E.7. The library supports CPU, GPU and IPU [§E.8] hardware to accelerate training. Further library optimisations are detailed in Appendix E.9.

## 4 EXPERIMENTS ON BASELINE MODELS

To demonstrate the capabilities of the `Graphium` library in a multi-task setting with thousands of labels, a set of standard baselines were run with simple hyperparameter sweeps using 3 popular GNNs, namely GCN (Kipf & Welling, 2017), GIN (Xu et al., 2019), and GINE (Hu et al., 2020a). A basic hyper-parameter sweep was conducted for each model and multiple random seeds used for initialisation to provide a performance baseline for future experiments to be evaluated against. In Appendix H, we report scaling law experiments on our proposed datasets and observed significantly improved results as models include more parameters.

In addition to the node features (and edge features in the case of GINE) provided with the datasets, we use (the first 8) eigenvalues and eigenvectors of the graph Laplacian as node-level PEs. Combining eigenvalues and eigenvectors and embedding them using an MLP offers a PE informed by the graph's spectral properties (Kreuzer et al., 2021). We further use random walk return probabilities (i.e., the probability of a random walk starting at a specific node and returning to the said node after $k \in \mathbb{N}$ steps), again, in combination with an MLP encoder (Dwivedi et al., 2021). This can indicate the presence of cycles within molecules, which can be informative in the context of some molecular tasks, e.g., the regression of the solubility (logP) score on the ZINC dataset (Dwivedi et al., 2020). We finally compare the results from the multi-task model to a single-task model. Details of hardware and computational resources can be found in Appendix G.

### 4.1 TOYMIX

**Experimental setting.** For all baseline models, the GNN module uses 4 GNN layers with a hidden size chosen to approximately balance the number of trainable parameters across models (reported in Tab. 2). For TOYMIX, the models have approximately 150k parameters. We train a model on each dataset individually and on the combined versions, with the results reported respectively in columns *Single Dataset* and *Multi Dataset*. For these cases we fix both the learning rate and total number of epochs (300) used. Importantly this means that the total number of training steps will be larger for the combined dataset since there are more unique molecules.

**Losses and metrics computation.** The multi-task loss is built by combining the loss function for each task. For the regression tasks, the models are trained with Mean Average Error (MAE) and evaluated using the Pearson correlation coefficient and the coefficient of determination $R^2$. For the classification tasks, the loss is the binary cross entropy (BCE) and the metrics are area under the receiver operating characteristic curve (AUROC) and the average precision (AP) for the precision-recall curve. The MAE and BCE are computed on the flattened tensor, but the other metrics are computed *per label* and then averaged. All "NaNs" are filtered out from the computation. Also, we calculate the Mean Average Error (MAE) on regression tasks using the *normalized* outputs and labels so the results are not skewed by the magnitude of the labels themselves. Normalisation constants can be found with the dataset and future users should use these values to ensure results can be compared.

**Comparability to results in previous works.** It should be noted that for the TOYMIX dataset, while the component datasets are well studied, the results present here are not directly comparable to prior work. Here QM9 has an extended list of 19 labels, that are normalized and aggregated, compared to 12 non-normalized labels typically reported in prior work. ZINC12K also uses 3 labels compared to the standard single label. Finally TOX21 uses a simple random split data split compared to the *scaffold split* used in prior literature.

Table 2: Results for GNN baselines on the proposed ToyMix dataset. We report performance metrics on the test set (mean $\pm$ std over 3 seeds) per dataset contained in ToyMix and for the dataset overall. The best score for each metric per dataset *across all three models* is in marked **green**, with the best result *per model* marked in **orange**.

| Dataset | Model | Single Dataset | | | Multi Dataset | | |
|---|---|---|---|---|---|---|---|
| | | MAE ↓ | Pearson ↑ | $R^2$ ↑ | MAE ↓ | Pearson ↑ | $R^2$ ↑ |
| QM9 | GCN | **.102** $\pm$ .0003 | **.958** $\pm$ .0007 | **.920** $\pm$ .002 | .119 $\pm$ .01 | .955 $\pm$ .001 | .915 $\pm$ .001 |
| | GIN | **.0976** $\pm$ .0006 | **.959** $\pm$ .0002 | **.922** $\pm$ .0004 | .117 $\pm$ .01 | .950 $\pm$ .002 | .908 $\pm$ .003 |
| | GINE | **.0959** $\pm$ .0002 | .955 $\pm$ .002 | **.918** $\pm$ .004 | .102 $\pm$ .01 | **.956** $\pm$ .0009 | **.918** $\pm$ .002 |
| Zinc12k | GCN | .348 $\pm$ .02 | .941 $\pm$ .002 | .863 $\pm$ .01 | **.226** $\pm$ .004 | **.973** $\pm$ .0005 | **.940** $\pm$ .003 |
| | GIN | .303 $\pm$ .007 | .950 $\pm$ .003 | .889 $\pm$ .003 | **.189** $\pm$ .004 | **.978** $\pm$ .006 | **.953** $\pm$ .002 |
| | GINE | .266 $\pm$ .02 | .961 $\pm$ .003 | .915 $\pm$ .01 | **.147** $\pm$ .009 | **.987** $\pm$ .001 | **.971** $\pm$ .003 |
| | | BCE ↓ | AUROC ↑ | AP ↑ | BCE ↓ | AUROC ↑ | AP ↑ |
| Tox21 | GCN | .202 $\pm$ .005 | .773 $\pm$ .006 | .334 $\pm$ .03 | **.176** $\pm$ .001 | **.850** $\pm$ .006 | **.446** $\pm$ .01 |
| | GIN | .200 $\pm$ .002 | .789 $\pm$ .009 | .350 $\pm$ .01 | **.176** $\pm$ .001 | **.841** $\pm$ .005 | **.454** $\pm$ .009 |
| | GINE | .201 $\pm$ .007 | .783 $\pm$ .007 | .345 $\pm$ .02 | **.177** $\pm$ .0008 | **.836** $\pm$ .004 | **.455** $\pm$ .008 |

**Results discussion.** ToyMix results can be found in Table 2. We compare 3 different models and find that across all of the different settings, GINE outperforms GIN, which in turn outperforms GCN. To investigate the impact of using the union of multiple datasets we compare the performance for single dataset training with those in a multi dataset context. Interestingly we find that for the smaller Zinc12k and Tox21 datasets there is a clear advantage to also training them with the additional data. This is striking as these tasks are not highly related and the chemical spaces don't intersect much. This highlights that learning in unison with other molecular tasks alone may be beneficial, especially knowing that quantum data like QM9 and chemoinformatics data like Zinc12k are much cheaper to obtain than Tox21 or other bioassay data. For the larger dataset QM9, which takes up nearly 87% of the molecules, however, we do see a minor reduction in the performance in the multi dataset context, which suggests that there is some degree of compromise when using this setup.

## 4.2 LargeMix

**Experimental setting.** In the case of LargeMix, we provide results for the same GNN baselines with 4 GNN layers and model sizes between 4M and 6M parameter. We present results of models trained for 200 epochs and average across 3 seeds. The remaining setup is the same as for ToyMix (Section 4.1), reporting metrics on the *Single Dataset* and *Multi Dataset* are the same.

**Results discussion.** LargeMix results can be found in Table 3. On the L100 datasets VCAP and MCF7, similar to ToyMix, we observe a clear advantage of our *Multi Dataset* approach compared to training on a single dataset. While not the case for the significantly larger PCQM4M and PCBA_1328 datasets. This again indicates the smaller datasets benefit more from the multi-dataset training approach. We believe that this is due to underfitting, as e.g., for PCBA_1328, a 10M parameter model is used to train on more than 3000 tasks jointly. Indeed, in Table 7, we observe that as we increase the model size, the results for PCQM4m_N4 and PCBA_1328 significantly improve.

## 4.3 UltraLarge

**Experimental setting.** For UltraLarge, we provide results for the same GNN baselines as for LargeMix. Each model is trained for 50 epochs and results are averaged over 3 seeds. The remaining setup is the same as for ToyMix (Section 4.1), reporting metrics on the *Single Dataset* and *Multi Dataset* using the same performance metrics. We further use the same models (in terms of size) as used for LargeMix. However due to resource constraints we have trained only on a 5% subset of the data.

**Results discussion.** UltraLarge results can be found in Table 4. Interestingly, on both graph- and node-level tasks we observe that there is no advantage of multi-tasking in terms of performance. However, we expect the major benefits of this dataset to be found when used at scale and when either fine-tuning to other quantum tasks or when used in conjunction with a more diverse array of datasets.

Table 3: Results for GNN baselines on the proposed LARGEMIX dataset. We report performance metrics on the test set (mean ± std over 3 seeds) per dataset contained in LARGEMIX and for the dataset overall. The best score for each metric per dataset *across all three models* is in marked **green**, with the best result *per model* marked in **orange**.

| Dataset | Model | Single Dataset | | | Multi Dataset | | |
|---|---|---|---|---|---|---|---|
| | | MAE ↓ | Pearson ↑ | $R^2$ ↑ | MAE ↓ | Pearson ↑ | $R^2$ ↑ |
| PCQM4M_G25 | GCN | .2362± .0003 | .8781± .0005 | .7803± .0006 | .2458± .0007 | .8701± .0002 | .7678± .0004 |
| | GIN | .2270± .0003 | .8854± .0004 | .7912± .0006 | .2352± .0006 | .8802± .0007 | .7827± .0005 |
| | GINE | .2223± .0007 | .8874± .0003 | .7949± .0001 | .2315± .0002 | .8823± .0002 | .7864± .0008 |
| PCQM4M_N4 | GCN | .2080± .0003 | .5497± .0010 | .2942± .0007 | .2040± .0001 | .4796± .0006 | .2185± .0002 |
| | GIN | .1912± .0027 | .6138± .0088 | .3688± .0116 | .1966± .0003 | .5198± .0008 | .2602± .0012 |
| | GINE | .1910± .0001 | .6127± .0003 | .3666± .0008 | .1941± .0003 | .5303± .0023 | .2701± .0034 |
| | | CE ↓ | AUROC ↑ | AP ↑ | CE ↓ | AUROC ↑ | AP ↑ |
| PCBA_1328 | GCN | .0316 ± .0000 | .7960 ± .0020 | .3368 ± .0027 | .0349± .0002 | .7661± .0031 | .2527± .0041 |
| | GIN | .0324 ± .0000 | .7941 ± .0018 | .3328 ± .0019 | .0342± .0001 | .7747± .0025 | .2650± .0020 |
| | GINE | .0320 ± .0001 | .7944 ± .0023 | .3337 ± .0027 | .0341± .0001 | .7737± .0007 | .2611± .0043 |
| L1000_VCAP | GCN | .1900 ± .0002 | .5788 ± .0034 | .3708 ± .0007 | .1872± .002 | .6362± .0012 | .4022± .0008 |
| | GIN | .1909 ± .0005 | .5734 ± .0029 | .3731 ± .0014 | .1870± .001 | .6351± .0014 | .4062± .0001 |
| | GINE | .1907 ± .0006 | .5708 ± .0079 | .3705 ± .0015 | .1862± .0007 | .6398± .0043 | .4068± .0023 |
| L1000_MCF7 | GCN | .1869 ± .0003 | .6123 ± .0051 | .3866 ± .0010 | .1863± .0011 | .6401± .0021 | .4194± .0004 |
| | GIN | .1862 ± .0003 | .6202± .0091 | .3876 ± .0017 | .1874± .0013 | .6367± .0066 | .4198± .0036 |
| | GINE | .1856 ± .0005 | .6166 ± .0017 | .3892± .0035 | .1873± .0009 | .6347 ± .0048 | .4177± .0024 |

Table 4: Results for GNN baselines on a 5% subset of the proposed ULTRALARGE dataset. We report performance metrics on the test set (mean ± std over 3 seeds) per dataset contained in ULTRALARGE and for the dataset overall. The best score for each metric per dataset *across all three models* is in marked **green**, with the best result *per model* marked in **orange**.

| Dataset | Model | Single Dataset | | | Multi Dataset | | |
|---|---|---|---|---|---|---|---|
| | | MAE ↓ | Pearson ↑ | $R^2$ ↑ | MAE ↓ | Pearson ↑ | $R^2$ ↑ |
| PM6_83M_G62 | GCN | .2606± .0011 | .9004± .0004 | .7997± .0009 | .2625± .0011 | .8993± .0005 | .7977± .0008 |
| | GIN | .2541± .0017 | .9053± .0013 | .8069± .0028 | .2578± .0014 | .8981± .007 | .8031± .0027 |
| | GINE | .2538± .0006 | .9057± .008 | .8078± .0013 | .2575± .0009 | .9043± .0002 | .8052± .0003 |
| PM6_83M_N7 | GCN | .5803± .0001 | .3372± .0003 | .1188± .0005 | .5971± .0002 | .3164± .0009 | .1019± .0011 |
| | GIN | .5729± .0002 | .3480± .0003 | .1271± .0003 | .5831± .0001 | .3314± .0004 | .114± .0008 |
| | GINE | .5721± .0004 | .3484± .0006 | .1262± .0006 | .5839± .0004 | .3294± .0003 | .1108± .0004 |

## 5 CONCLUSION

In this work, we curated a novel collection of molecular datasets with an unprecedented number of data points for supervised learning, thereby significantly enhancing the resources available for research in the field of drug discovery. These datasets are naturally multi-task and multi-level, posing unique challenges for machine learning models. To facilitate the study of these complex datasets, we presented the `Graphium` library, a framework designed to process and efficiently load large-scale molecular data and perform multi-task learning across various task levels. The library takes advantage of the unique features and properties presented by our datasets and build towards the training of a large foundational model for molecular learning.

Furthermore, we present a number of baseline results on our proposed datasets and show that performance on small biological datasets benefits from being trained on large amounts of quantum data. We hypothesise that this improved generalization will help foundation models perform well when fine-tuned to the low-resource downstream tasks commonly encountered in Drug Discovery.

REPRODUCIBILITY

In line with ICLR's commitment to promoting reproducibility in research, we have taken diligent steps to ensure the reproducibility of our work presented in this paper. The `Graphium` library code can be accessed on Github: `https://github.com/datamol-io/graphium`. This code allows fellow researchers to reproduce and build upon our experimental results.

All datasets from TOYMIX, LARGEMIX and ultra-large can be found below. Datasets are provided either as *.csv* for smaller datasets or *.parquet* for larger datasets. Splits are provided as *.pt* files, meant to be read via *torch.load*, and containing a dictionary with keys "train", "val", "test" and "test-seen", each containing a list of indices corresponding to the split.

Download links for the *ToyMix*, *LargeMix* and *UltraLarge* datasets are found in Zenodo Part 1(`https://zenodo.org/records/7998401`) and Part 2(`https://zenodo.org/records/8370547`).

ETHICS STATEMENT

We confirm that we adhere to the ICLR Code of Ethics as stated here. This research primarily involves the analysis and prediction of molecular data using machine learning techniques. We have taken ethical considerations into account at various stages of our work. Our experiments do not involve human subjects, and we have not accessed any sensitive or private information. The licenses for the datasets contributed in this work are discussed in Appendix B.2.

ACKNOWLEDGMENTS

This work was conducted by Shenyang Huang, Gabriela Moisescu-Pareja, Oleksandr Dymov, and Frederik Wenkel during an internship at Valence Labs. Shenyang Huang is funded by NSERC PGS D Award and FRQNT Doctoral Award. Guillaume Rabusseau and Reihaneh Rabbany are supported by the CIFAR AI chair program. Frederik Wenkel is partially funded by the Fin-ML CREATE graduate studies scholarship for PhD, the J.A. DeSève scholarship for PhD and Guy Wolf's research funds. Guy Wolf is funded by IVADO (Institut de valorisation des données) grant PRF-2019-3583139727, FRQNT (Fonds de recherche du Québec - Nature et technologies) grant 299376, NIH (National Institutes of Health) - NIGMS grant R01GM135929 and the Canada CIFAR AI Chair. Christopher Morris and Luis Müller are partially funded by a DFG Emmy Noether grant (468502433) and RWTH Junior Principal Investigator Fellowship under Germany's Excellence Strategy. Michał Koziarski is funded by the Genetech-Mila partnership. Jian Tang is supported by Twitter, Intel, the Natural Sciences and Engineering Research Council (NSERC) Discovery Grant, the Canada CIFAR AI Chair Program, Samsung Electronics Co., Ltd., Amazon Faculty Research Award, Tencent AI Lab Rhino-Bird Gift Fund, an NRC Collaborative R&D Project (AI4D-CORE-06) as well as the IVADO Fundamental Research Project grant PRF-2019-3583139727. The content provided here is solely the responsibility of the authors and does not necessarily represent the official views of the funding agencies.

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

## A    LIMITATIONS, FUTURE WORK AND SOCIETAL IMPACTS

**Limitations and Future Work** Our proposed datasets currently do not contain edge- or nodepair-level tasks, which we plan to include in the future. Also, our benchmark results are limited to a few baseline models, but we will continue to run experiments and ablations and eventually include more positional encodings and graph Transformers. Further, large-scale multi-task learning on millions of molecular data may pose a limitation in situations where extensive computing resources are unavailable. We will continue to work on optimization, including chip-specific accelerations, to help reduce the time and cost of training. On that same note, the ultra-large PM6_83M dataset is so large that using it in the LargeMix might be challenging since in the current `Graphium` framework, 95% of the molecules in any batch will come from that specific dataset, and under-sampling might be required. Finally, since the capability of a model trained on multiple tasks to adapt and perform well on new tasks is a crucial indicator of its robustness, we aim to investigate our multi-task framework's ability to transfer to new tasks.

**Societal Impact** Releasing our datasets and the `Graphium` library may not have immediate direct societal impacts. However, it is crucial to acknowledge the potential implications that arise when providing access to a foundation model for molecular graphs. One concerning possibility is the misuse of this technology for the development of chemical weapons, toxins, or unregulated drugs. To address these potential risks, we are committed to implementing robust mitigation strategies. Central to our approach is the active promotion of beneficial applications, particularly in the fields of material and drug discovery. By highlighting the positive utilization of this technology, we aim to channel its potential toward scientific advancements that contribute to societal well-being.

## B    DATASET DOCUMENTATION AND INTENDED USES

### B.1    INTENDED USE

The datasets are intended to be used in an academic setting for training molecular GNNs with orders of magnitude more parameters than current large models. Further, the TOYMIX and LARGEMIX datasets are intended to be used in a multi-task setting, meaning that a single model should be trained to predict them simultaneously. We invite users to use the `Graphium` library and contribute to it to simplify the development of state-of-the-art GNNs on multi-task and multi-label data.

### B.2    DATASET LICENSES

The licenses of the existing datasets used in this work is as follows:

- **QM9**: Creative Commons Attribution Non-Commercial ShareAlike 4.0
- **Tox21**: Attribution 4.0 International (CC BY 4.0)
- **Zinc12k**: MIT license
- **L1000**: free to use by all
- **PCQM4M** and **PM6_83M**: Creative Commons Attribution 4.0 International license.
- **PCBA1328**: Open access

All datasets provided in this work are licensed under the Attribution Non-Commercial ShareAlike 4.0 International (CC BY-NC-SA 4.0) license. We chose this license because some of the original datasets have this license and we provide our datasets with the same level of access.

### B.3    MOLECULAR PROPERTIES

For the PM6_83M dataset, we added many chemical properties using `rdkit`, which are predicted together with the quantum properties. This enables the trained model to learn a shared embedding between chemical and quantum properties.

The computed chemical properties are:

- MW: molecular weight.

- fsp3: fraction of sp3 carbon atoms

- TPSA: topological polar surface area

- QED: quantitative estimation of drug-likeness

- clogp: partition coefficient for n-octanol/water (measure of lipophilicity)

- SAS: synthetic accessibility score

- n_lipinski_hba: number of hydrogen bond acceptors

- n_lipinski_hbd: number of hydrogen bond donors

- n_rings: number of ring structures

- n_hetero_atoms: number of heteroatoms (non hydrogen or carbon atoms)

- n_heavy_atoms: number of non-hydrogen atoms

- n_rotatable_bonds: number of rotatable bonds

- n_radical_electrons: number of radical electrons

- n_aliphatic_carbocycles: number of aliphatic carbocycles

- n_aliphatic_heterocyles: number of aliphatic heterocycles

- n_aliphatic_rings: number of aliphatic rings

- n_aromatic_carbocycles: number of aromatic carbocycles

- n_aromatic_heterocyles: number of aromatic heterocycles

- n_aromatic_rings: number of aromatic rings

- n_saturated_carbocycles: number of saturated carbocycles

- n_saturated_heterocyles: number of saturated heterocycles

- n_saturated_rings: number of saturated rings

## C  VISUALIZING THE DATASETS

### C.1  A FEW MOLECULES

We randomly sample 5 molecules from each dataset and visualize them in Figure 2. The PM6_83M dataset is omitted since it includes the PCQM4M_G25_N4 and the PCBA_1328.

### C.2  TOYMIX VISUALISATION

The chemical spaces of the dataset mixes are visualised in Figure 3 showing the overlap between dataset components. In the TOYMIX, we observe that ZINC12k and Tox21 share part of the chemical space, although Tox21 is more diverse, but QM9 is very different from both of them. In the LARGEMIX, we observe that there is a strong intersection between the datasets at the center of the plot, but that the MCF7 and VCAP, despite being smaller, cover larger regions of the chemical space, justifying their usefulness for pre-training foundation models of chemistry.

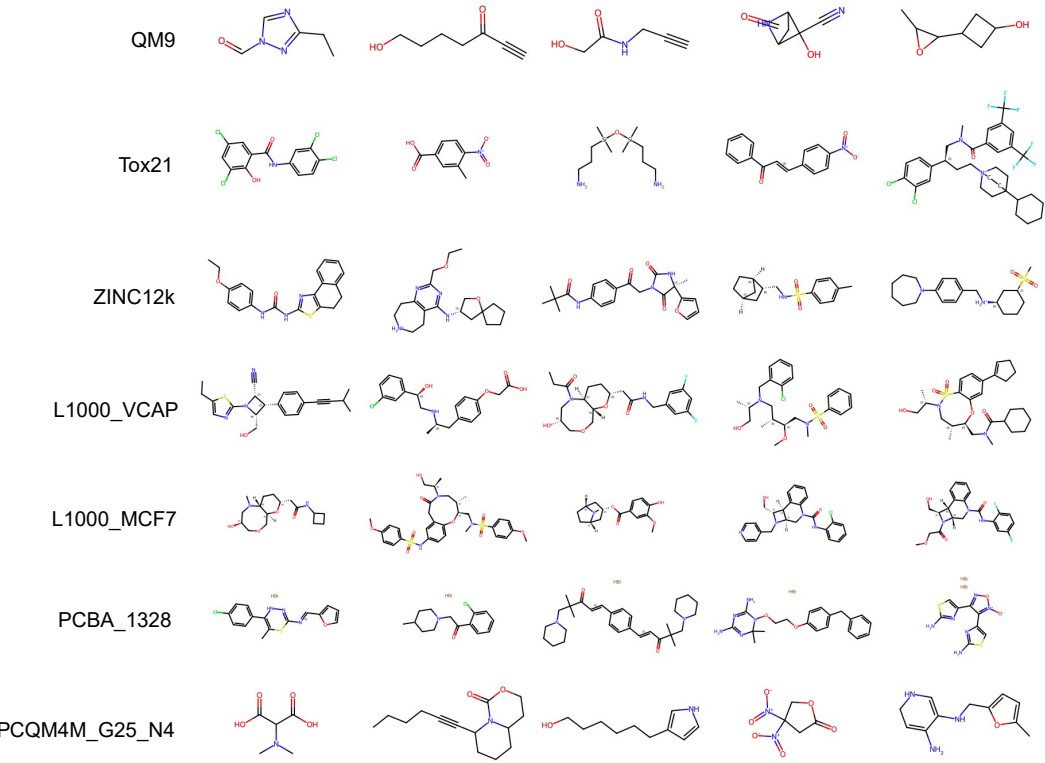

Figure 2: Visualizing random samples from the datasets.

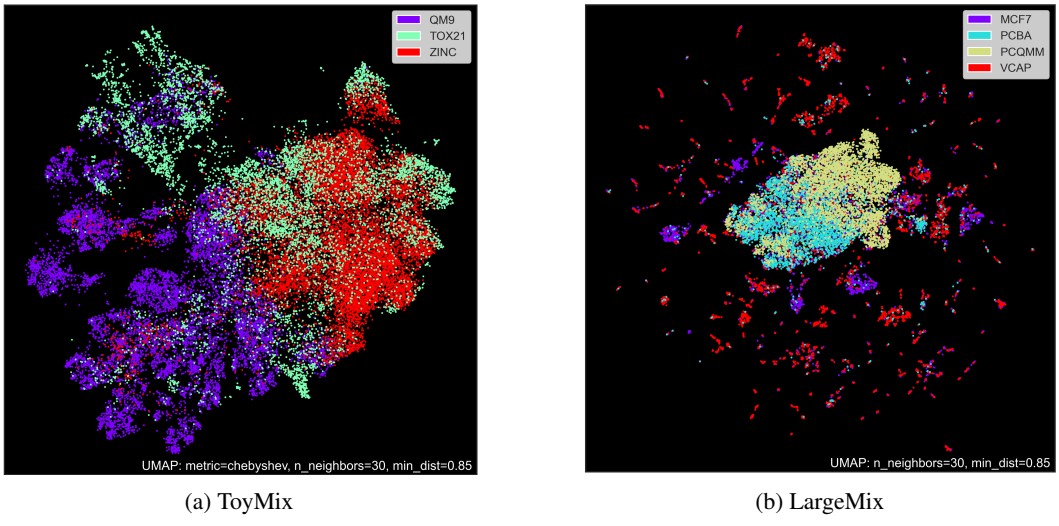

(a) ToyMix                                    (b) LargeMix

Figure 3: UMAP visualizations for the dataset mixes, computed with the ECFP molecular fingerprints, showing the chemical spaces of the datasets.

| Mix | Graph labels | Node labels | Total |
|-----|-------------|-------------|-------|
| TOYMIX | 2.6M | - | 2.6M |
| LARGEMIX | 344M | 197.7M | 541.7M |
| ULTRALARGE | 4B | 8.5B | 12.5B |
| Total | 4.3B | 8.7B | **13.04B** |

Table 6: Overview over the number of graph and node labels by mix.

## C.3 DATASET STATISTICS

Table 5: General statistics about the datasets, where G. / N. respectively indicate graph-level / node-level tasks, and where the sparsity indicates the % of missing data. h. atoms and h. bonds denote respectively the heavy atoms, and the bonds between 2 heavy atoms. For data type, Q. indicates it is Quantum, C. means it is Computed, B. means it is Bio-assays and T. means it is Transcriptomics. For task, C., R., and RC. denote *classification*, *regression* and *ranked classification* respectively.

| Mix | TOYMIX | | | LARGEMIX | | | | ULTRALARGE |
|-----|------|--------|-------|------------------|-----------|------------|------------|---------|
| Dataset | QM9 | ZINC12K | TOX21 | PCQM4M_G25_N4 | PCBA_1328 | L1000_VCAP | L1000_MCF7 | PM6_83M |
| Type | Q. | C. | B. | Q. | B. | T. | T. | Q. |
| Task | R. | R. | C. | R. | C | RC. | RC. | R. |
| # mols | 133.8k | 12k | 7.8k | 3.8M | 1.5M | 15.2k | 11.6k | 83.7M |
| # G. labels | 19 | 3 | 12 | 25 | 1,328 | 978 | 978 | 62 |
| # G. data | 2.5M | 36k | 74.9k | 93.4M | 224.3M | 14.8k | 11.3k | 4.00B |
| % G. sparsity | 0 | 0 | 20.2 | 1.9 | 89.2 | 0 | 0 | 22.9 |
| # N. labels | - | - | - | 4 | - | - | - | 7 |
| # N. data | - | - | - | 197.7M | - | - | - | 8.51B |
| % N. sparsity | - | - | - | 7.9 | - | - | - | 40.8 |
| # h. atoms | 1.18M | 278k | 146k | 53.7M | 41.4M | 509k | 362k | 2.06B |
| # h. bonds | 1.26M | 299k | 151k | 55.2M | 44.8M | 550k | 395k | 2.20B |

We analyze the different datasets by using a histogram over relevant chemical properties in Figure 4. There, we observe that the quantum datasets (QM9 and PCQM4M) contain much smaller molecules than the other datasets. This could hinder their ability to produce an embedding that generalizes well across the chemical space. However, the semi-empirical dataset PM6_83M, has very similar distribution to biologically relevant datasets, which makes it ideal for pre-training a foundation model.

## D DEEPER DIVE INTO THE LARGEMIX AND ULTRALARGE DATASETS

In this section, we go into further details to explain what labels are present in the datasets, and how they compare to the current literature.

### D.1 PCQM4M_G25_N4

As mentioned in section 2.2, this dataset comes from the same data source as the OGBG-PCQM4M, namely the PubChemQC project (Hu et al., 2021b), meaning that it comprises of the same molecules.

However, this dataset differs from OGBG-PCQM4M since additional properties are included, and the 3D descriptors are provided as labels. This implies that we can't use 3D structure as input, as done by OGB, since it is contained in the labels and would lead to data leaks. Further, the OGB challenge only provides labels for HOMO-LUMO gap predictions alongside the 3D structures for the training set. The reasoning behind OGB's choice of including the 3D conformation only in the training set comes from their desire that a model learns to implicitly embed it when predicting its associated HOMO-LUMO gap. However, since we work in a multi-task setting, it is harder for us to use the same technique. For example, suppose that we are also predicting the binding affinity as another task, then the 3D conformation responsible for this affinity is not necessarily the same as the one provided in the PCQM4M dataset. Therefore, our choice of including the 3D conformation in the labels will allow the model to better associate different conformation to different tasks.

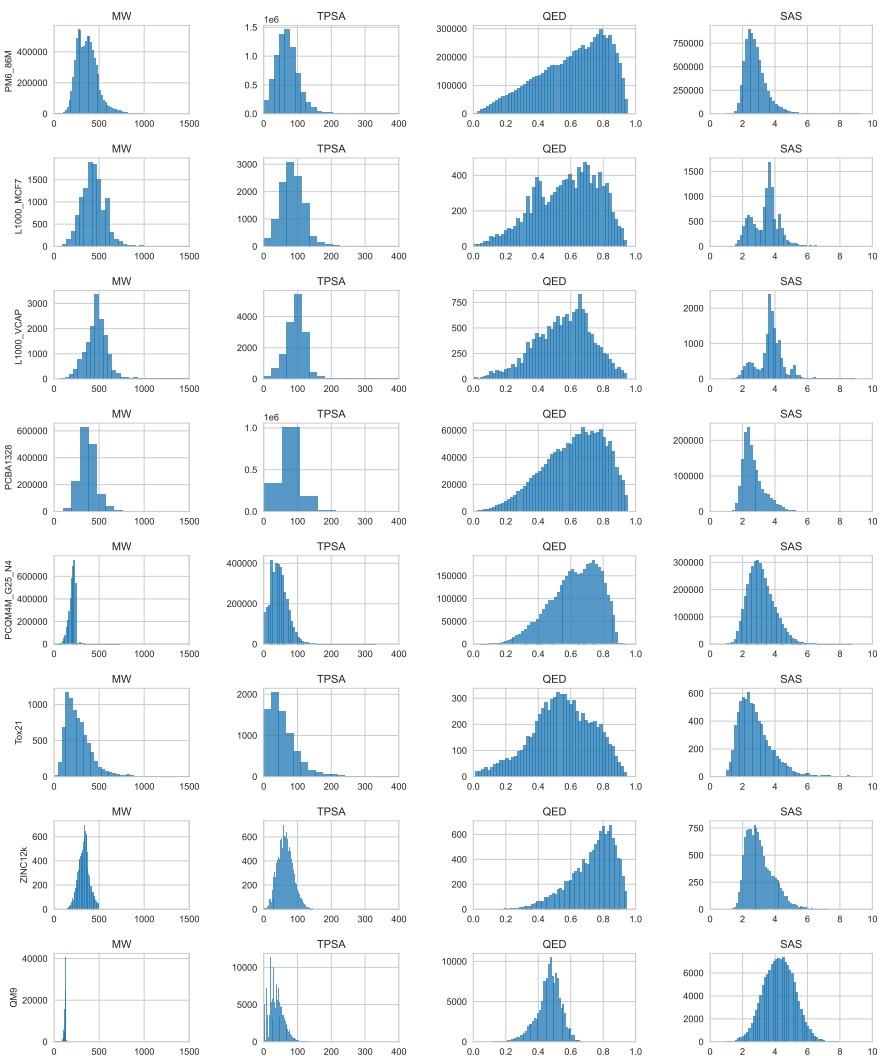

Figure 4: Histograms of the datasets for various properties relevant to chemists. These include molecular weights (MW), topological polar surface area (TPSA), quantitative estimation of drug-likeness (QED), and synthetic accessibility score (SAS). MW is the sum total of the atomic weights for all atoms in a molecule. TPSA is the surface sum over all polar atoms (like oxygen and nitrogen) and estimates the surface area of a molecule that is available for interactions with polar groups in biological systems. QED measures how drug-like a given compound is and is based on various molecular properties like MW, lipophilicity, hydrogen bond donors/acceptors, TPSA and the presence of certain functional groups. SAS is a metric to assess the feasability of synthesizing a given compound and is calculated based on several descriptors, such as size, complexity and stereochemistry. For the PM6 dataset, we calculated and plotted the descriptors for ∼10M molecules.

The DFT properties that we predict at the graph level are the following:

- Alpha HOMO
- Alpha HOMO-LUMO gap
- Beta HOMO
- Beta HOMO
- Total energy

It should be noted here that only the Alpha HOMO-LUMO gap is provided in the original OGBG-PCQM4M dataset. For most molecules, the number of labels are duplicated for time-dependent DFT (TDDFT), which allows to also model the excitation states of the molecule.

Using the 3D conformation optimized by the DFT, we further computed the following 3D descriptors as labels:

- Spherocity
- Plane of best fit
- Principal length *
- Mass inertia *
- Valence inertia *
- Mulliken charges inertia *
- Lowdin charges inertia *

Values denoted with "3D" means that the descriptors are computed alongside the 3 main axes of the molecule. The descriptors are inspired by existing 3D descriptors such as WHIM (Todeschini & Gramatica, 1997).

Finally, the node property predictions are the atomic charges for both DFT and TDDFT when available.

- Mulliken charges
- Lowdin charges

## D.2    L1000 VCAP AND MCF7

As discussed in section 2.2, the LINCS L1000 is a database of high-throughput transcriptomics that screened more than 30,000 perturbations on a set of 978 landmark genes (Subramanian et al., 2017) from multiple cell lines. VCAP and MCF7 are, respectively, prostate cancer and human breast cancer cell lines, with more details given below.

**MCF7** is a breast cancer cell line possessing drug-discovery relevant characteristics of differentiated mammary epithelium, such as estrogen receptors and dome formation3. Isolated in 1970 from a woman with metastatic adenocarcinoma of the breast, it became widely used as a model for breast cancer research and drug testing.

**VCAP** is a prostate cancer cell line exhibiting drug-discovery relevant features of advanced prostate cancer, such as androgen receptor signaling and bone metastasis. Isolated in 1997 from a man with hormone refractory prostate cancer, it became widely used as a model for prostate cancer research and drug testing.

Many perturbagens are used, but in our case, we focus only on small molecular perturbagens. Due to the high noise in the assays, we convert the task to a multi-class classification, with one class having a perturbation z-score $< -2$ and another $> 2$, and the neutral class with a perturbation in the range $[-2, 2]$.

There is a lot of noise in the L1000 regression dataset which can make it difficult to give accurate predictions, therefore, the floating-point dataset is converted into a binary classification task. By applying a threshold, we make the signal easier to distinguish from noise. While this can occasionally lead to mislabeled elements, our experience demonstrated that a classification task performed more reliably then a regression task.

### D.3 PCBA_1328

This dataset is gathered in a similar way to the OGB-PCBA (Hu et al., 2020b), by scraping some of the largest assays from Pubchem, but at a much larger scale.

Due to the unknown noise present in each assay, the variability between repeated experiments, and the difficulty associated with properly curating the datasets, we followed the same process as the OGB-PCBA data. Namely, we considered only the assays with author's labeling that mention "Active" for the positive class and "Inactive" for the negative class. All other labels were considered as missing values, since they are often about experimental errors.

As described in Appendix D.1, this dataset has several key differences to the original OGB-PCQM4M dataset. First, it covers 1328 assays, a 10-fold increase, and 1.5M unique molecules, a 3-fold increase. This is due to our less stringent process of selecting the assays. Indeed, every assay prior to 2022 with more than 6000 datapoints and more than 20 points in the positive class is selected. This implies that other popular datasets such as Tox21 (Huang et al., 2021) and OGB-HIV (Hu et al., 2020b) are also included in the dataset. Further, all 561 protein assays from the high-throughput fingerprints (Laufkötter et al., 2019) are present, with a prior demonstration of the meaningfulness of their molecular representation.

We cannot name all the assays present since the process was automated. We further omitted assays from 2022 and 2023 to allow future work to use them for benchmarking purposes.

### D.4 PM6_83M

As discussed in section 2.3, this dataset contains PM6 semi-empirical computation of quantum properties, a faster but less accurate computation than DFT's (Nakata & Shimazaki, 2017; Nakata et al., 2020). Covering 83M unique molecules, 62 graph-level tasks, and 7 node-level tasks, this is the largest dataset available for training 2D-GNNs regarding the number of unique molecules. In total, there are 221M PM6 computations (Nakata et al., 2020).

Just like the PCQM4M_G25_N4 dataset contains an excited state, this one contains four different molecular states, namely "S0" for the ground state, "T0" for the lowest energy triplet excited state, "cation" for the positively charged state, and "anion" for the negatively charged state.

The quantum properties that we predict at the graph level are the following. For many molecules, the number of labels are multiplied by 4 to account for the S0, T0, cation, and anion states of the molecule.

- Alpha HOMO
- Alpha HOMO-LUMO gap
- Beta HOMO
- Total energy

It should be noted here that only the Alpha HOMO-LUMO gap is provided in the original OGBG-PCQM4M dataset.

Using the 3D conformation optimized by PM6, we further computed the following 3D descriptors as labels. Values denoted with "3D" means that the descriptors are computed alongside the 3 main axes of the molecule. We compute less descriptors than the PCQM4M to avoid having too many properties, especially considering that there are now 4 excited states and a lot more molecules.

- Spherocity
- Plane of best fit
- Principal length *

Then, the node property predictions are the atomic charges and spins for all S0, T0, anion and cation states.

- Mulliken charges

- Electronic spin

Finally, all 22 properties from B.3 are also added as labels.

## D.5 OTHER RELEVANT DATASETS IN THE LITERATURE

There are many other large datasets in the literature of molecular discovery. In this section, we give a brief description of a few of these datasets and how they are different or not suited to our current problem.

First, there are a few large molecular datasets aimed towards material discovery. For these, both the chemical space and the set of tasks / molecular properties don't intersect. Such examples are **NablaDFT** (Khrabrov et al., 2022) and **OpenCatalyst** (Chanussot et al., 2021), with the former being focused on trajectories.

Then, there are other datasets more related to drug discovery, but whose goals are to understand molecular dynamics within pockets **MISATO** (Siebenmorgen et al., 2023) or to learn more accurate force-fields **Spice** (Eastman et al., 2023). Both of these tasks are not suited for our current desire of training a large 2D-GNNs that work on the graph alone, although they could be suited for 3D-GNNs.

Finally, datasets such as **GEOM** (Axelrod & Gomez-Bombarelli, 2022) were shown to be very useful for pre-training GNNs, however they require dedicated architectures and their performance remain limited (Stärk et al., 2022; Liu et al., 2023a).

## D.6 RELATION BETWEEN THE DATASETS

There are very little direct relations between the datasets as they cover a wide variety of tasks. However, our empirical results show that running in a multi-task setting improves the results compared to single-task experiments. This is because there are some relationships between them, and understanding one can lead to a better understanding of the other.

First, the quantum simulation datasets, namely PCQM4M_G25_N4 and PM6_83M, contain information about the inner workings of molecules, how changing atoms affect the charges, the polarity, the 3D structure, the energy and excitation energies, the electronic densities, etc. All these are relevant when considering binding affinity to a protein, thus they are expected to help improve generalization on the binding assays that make up the majority of PCBA_1328.

We further argue that many ADME (absorption, distribution, metabolism, and excretion) properties depend on the physics and shape of the molecules, and they are very relevant to both binding and cell assays. Hence, learning on the quantum datasets is likely to benefit the other ones.

Second, learning the binding affinity alongside other assays in PCBA_1328 can lead to an improvement in transcriptomics predictions. Indeed, molecules mostly affect the cells via inhibition or activation of proteins, and this effect will often be visible via the changes in gene expression. Hence, we expect an improvement of the L1000 predictions when combining it to PCBA_1328, and by induction, to the other quantum datasets.

Third, learning on L1000 transcriptomic is expected to lead to an improvement in the prediction of biological properties of molecules since the studied cell-lines are simpler models of the human biology and diseases. Hence, despite not providing such dataset in the current work, we expect that fine-tuning pre-trained models on such tasks will be beneficial.

## E FURTHER DETAILS ON THE GRAPHIUM LIBRARY

The following section provides further details about the `Graphium` library.

### E.1 MULTI-LEVEL, MULTI-TASK AND MULTI-LABEL LEARNING IN GRAPHIUM

`Graphium` provides an efficient mechanism for loading and processing multi-task datasets and handling tasks in parallel to enhance computational efficiency significantly. It supports multiple tasks, including node, edge, nodepair, and graph-level prediction tasks. Additionally, the library

incorporates multi-level positional encodings; see Appendix E.3. Such encodings enhance the model's capacity to capture and learn from the inherent hierarchical relationships within the molecular data (Rampášek et al., 2022), providing a unified platform for handling diverse data requirements. In `Graphium`, we propose a multi-task, multi-level model framework as illustrated in Figure 5. After pooling, the architecture utilizes GNN layers to extract node, edge, nodepair, and graph-level features. The resulting features are subsequently fed into task-level specific MLPs, which, in turn, are fed into task-specific heads, generating predictions at node, edge, nodepair, and graph-level tasks. This multi-level prediction strategy accounts for complex hierarchical relationships within molecular data, allowing the model to perform simultaneous learning tasks at different scales.

**Multi-level Learning:** In `Graphium`, The term *multi-level* refers to a model's capability to learn and output representations at various levels of the graph. `Graphium` supports four levels: node, edge, graph, and node-pair. To simplify this, let's use molecular networks as an example.

On the node level, the task is to predit a property of the individual nodes within a molecular graph. Each node often represents an atom; hence, a node-level task might involve learning representations of an atom's properties or its role within a particular molecular graph. The edge level focuses the relationship between connected nodes in a graph. In molecular networks, the edge level aims to learn the embeddings of the type of bond (e.g. single, double or aromatic) or the relationship between atoms or proteins.

On the graph level, a property of the entire graph is considered. In molecular networks, this involves learning the representations of molecular properties, like its solubility or stability.

Lastly, the node-pair levels focuses on the interactions between arbitrary pairs of nodes. Examples including predicting the distance between all pairs of atoms in molecular graphs.

In `Graphium`, task at each level starts with data featurization, then to the GNN layers (e.g. MPNN or Transformer layers), and lastly level-specific MLPs to get each level's embedding. The graph level shares a similar path but uses pooled graph-level features as shown in figure 5. Data featurization such as positional encodings, architecture and metrics, and any configurations that are related to the levels are easily managed via configuration files.

**Multi-task Learning:** `Graphium` supports multi-task learning where different tasks are learned jointly on the same dataset potentially at different levels. Each level can also support multiple tasks. Each task has its specific task output head (usually constructed as a MLP), allowing the model to handle diverse tasks. For instance, at the node level, tasks could range from "node task 1" to "node task N", with similar structures available for other levels. Beyond working on tasks within a single level, `Graphium` can concurrently optimize tasks across different levels. For example, it can be trained to predict an atom's property (a node-level task) while also determining the type of bond (an edge-level task). Managing the losses and metrics (see section 4.1) for each task is straightforward, with everything organized within a single configuration file.

**Multi-label Learning:** Multi-label refers to scenarios where each sample can belong to more than one class. Unlike traditional classification where each instance is associated with a singular label, in multi-label learning, each molecular instance can be associated with a set of labels. For example, a particular molecule might simultaneously be therapeutic and toxic, or have both antibacterial and antifungal properties. `Graphium` accommodates various tasks, including binary label, multi-label, regression, and multi-class classifications.

## E.2 MODELING

`Graphium` supports straightforward hyper-parameter sweeps, enabling users to fine-tune their models efficiently, improving performance across multiple tasks. Each task is tracked independently using its metrics, ensuring a detailed understanding of the model's performance across different tasks. The library supports a variety of GNNs, including 2D, 3D, and graph transformers, providing a seamless user experience and facilitating easy switching between different model architectures.

Expanding `Graphium` to support more expressive architectures, including transformer models, is a critical step in advancing foundation models for drug discovery. The transformer architecture is known for its capacity to capture complex, long-range dependencies and relationships within the data (Rampášek et al., 2022). In molecular graphs, such relationships often contain critical information

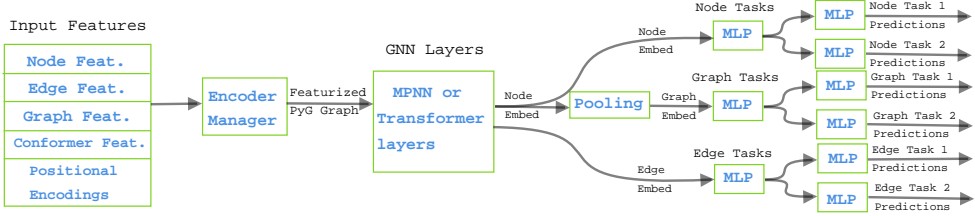

Figure 5: **Flow chart of multi-level multi-task molecular learning library, `Graphium`.** The `Graphium` library offers a comprehensive pipeline for molecular learning, consisting of 1) advanced molecular featurization, 2) positional encodings, 3) state-of-the-art GNN and graph transformer layers, 4) multi-level MLPs, and 5) multi-task learning through task-specific MLPs at the node, edge, and graph levels.

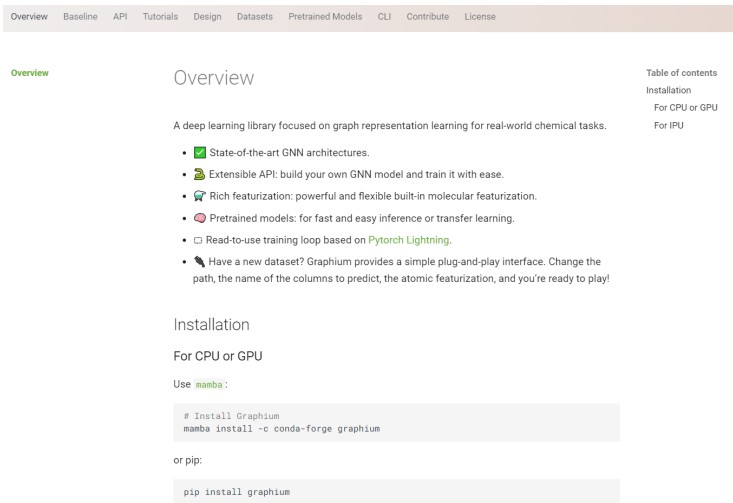

Figure 6: Documentation website layout for `Graphium` including API references, tutorials, dataset details and more.

about a molecule's properties. Additionally, due to their self-attention mechanism, transformer models can provide better interpretability, a component much needed in drug discovery. Thus, by supporting more expressive architectures, the library advances the development of foundation models that are not only highly accurate but also interpretable, fostering a broader adoption in the scientific community.

To help users understand better the functionalities and APIs in `Graphium`, we constructed an extensive documentation website for `Graphium` where API references, tutorials, dataset details and more can be found. This website will be made publicly available and as an example, we include an overlook of the website in Figure 6.

### E.3 POSITIONAL ENCODINGS

`Graphium` integrates multi-level positional encodings (PEs) at the node, edge, node-pair, and graph levels into a customizable molecular multi-level and multi-task graph learning framework. It enables flexible use of various PE combinations, easily managed through a compact configuration file. The library introduces novel *position transfer functions* that facilitate information exchange across positional levels. The most prevalent PEs are on the node level and provide nodes with an understanding of their positions within a (sub)graph. We further encode relationships between pairs of nodes via edge and nodepair PEs and global graph structure through graph PEs. With a modular

design, `Graphium` supports easy integration of new PEs at all levels, enhancing its ability to handle complex multi-task learning scenarios effectively.

### E.4 Label Normalization

Label normalization is a well-known technique that improves the regression ability of a model by making the labels easier to learn. In a setting with thousands of labels coming from different datasets, the losses associated with each label can be very imbalanced. To counteract that problem, `Graphium` natively supports multiple normalization schemes, including min-max normalization and z-score standardization.

### E.5 Ranked Classification (Hybrid Regression + Classification Loss)

For the L1000 datasets, we converted the data from regression classification by binning it into 5 categories. To avoid losing the ranking of the classes, we use a hybrid between cross-entropy and MSE loss. Formally, we define this hybrid loss below, with $C$ denoting the number of classes, $\mathbf{x} \in [0, 1]^C$ denoting the prediction, $y \in \{0, 1, ..., C - 1\}$ as the target, and $\alpha$ as the weight of the CE loss, as

$$\mathcal{L}(\mathbf{x}, y) = \alpha \Big( - \sum_{i=0}^{C} \mathbb{1}(i = y) \log(x_i) \Big) + (1 - \alpha) \Big( \sum_{i=0}^{C} (x_i \cdot i) - y \Big)^2. \tag{1}$$

### E.6 Handling of Missing Data

An essential component of our multi-tasking paradigm is processing graph data from multiple sources in the same batch. However, graphs in a dataset $\mathcal{D}_1$ will likely not appear in another dataset $\mathcal{D}_2$, resulting in missing labels. In addition, many of the proposed datasets are sparse. Hence, the per-task loss must be computed as if the batch consisted exclusively of predictions for which the targets are available.

As a result, `Graphium` supports different strategies to ignore the missing data or NaNs. These include replacing NaNs with a constant number or filtering out NaNs. However, strategies involving filtering do not work when the code is compiled, which in turn, is necessary for faster training. In these cases, we weigh the missing targets with $0$ to compute the desired per-task losses in a differentiable way, thereby allowing parallel graph processing from different sources and with missing targets. Specifically, for a task with predictions $\mathbf{x} \in \mathbb{R}^T$ and vector of targets $\mathbf{y} \in \mathbb{R}^T$, we compute

$$\mathcal{L}(\mathbf{x}, \mathbf{y}) \cdot \frac{T}{T - N_{\text{nan}}},$$

as the final task loss, where $\mathcal{L}$ denotes the per-task loss, $T$ denotes the number of targets and $N_{\text{nan}} < T$ is the number of `nan` targets.

### E.7 Support for Scalable Hyper-parameter Tuning With μP

To mitigate expensive hyper-parameter tuning, `Graphium` deeply integrates with $\mu$P, a pre-training paradigm that allows for tuning parameters on a small model, which can be subsequently transferred to a larger version with no or minimal additional tuning required (Yang et al., 2022). We support $\mu$P out of the box, thus enabling models to scale to billions of parameters efficiently. To the best of our knowledge, this is the first work enabling $\mu$P for GNNs.

### E.8 IPU Support

To maximize the throughput of training and inference in `Graphium`, we natively support Graphcore IPUs (Jia et al., 2019). The most recent BOW IPUs have 350 TFLOPs of `float16` arithmetic at peak performance and nearly 900MB of high bandwidth on-chip SRAM distributed across 1,472 individual compute cores which can execute programs independently of all other cores. This has the general benefit of allowing model weights and activations to be stored directly on SRAM but also provides a highly flexible, fine-grained fabric for compute parallelization. Additionally, this has specific benefits for GNN-like model structures as it allows to accelerate sparse communication operations like gathers

and scatters (Helal et al., 2022) and efficiently parallelizes complex combinations of smaller matrix multiplications that can be found in SOTA GNNs (Masters et al., 2023). This makes it an ideal hardware platform for the model types targeted by `Graphium`.

### E.9 LIBRARY OPTIMIZATION

`Graphium` is optimized for large-scale molecular datasets and models, allowing for fast processing and loading, and is compatible with IPU accelerations while supporting traditional CPU and GPU resources; see below for details.

**Dataloading Optimizations** are implemented to handle large-scale datasets efficiently, molecules are processed in parallel to apply a set of user-defined featurizers. Additional performance is obtained using a batched-parallel implementation, where each worker processes a batch of molecules, typically on the order of thousands. As the scale of the datasets grows, keeping the data in memory becomes a challenge, particularly as memory usage can grow linearly with the number of workers. To avoid these issues, in `Graphium`, each molecule is stored as a separate file on the disk. During data-loading, each worker receives the number of the molecule in the dataset to load, which directly corresponds to the filename to process, massively reducing data serialization and memory usage. Furthermore, storing molecules on disk provides an opportunity to skip pre-processing the dataset for every training run.

**PE Caching** We implement caching of intermediate computations related to the PE derivations (e.g., eigendecomposition of the graph Laplacian and higher-order diffusion operations) that are relevant for the construction of several PEs. This is especially beneficial when a large selection of PEs is used in a model. Further, we can thereby use positional information across several positional levels (via the position transfer functions proposed in Appendix E.3) with minimal additional computations required.

**IPU-specific Optimizations** are supported by `Graphium`, such as multiple device iterations (performing multiple training steps on the device, reducing synchronization between the host and the accelerator), buffer prefetch (placing multiple batches on the device ahead of time), IO tiles (dedicating a specified number of IPU tiles to data transfer), and loading data asynchronously on the host while the IPU is processing. The Poplar software stack used to run programs on the IPU requires that the shapes of tensors be fixed at compile time, so a packing strategy is also applied to the input data to minimize the padding data required. The library also supports pipelining large models over multiple processors.

**Low Precision Training** in both mixed precision and true 16 bit floating point precision support in `Graphium` allows reducing training time and memory requirements.

## F  TOKEN COUNT AND COMPARISON TO GPT-2

Here we compare our proposed datasets with datasets used to pre-train foundational models in language, in particular GPT-2, pre-trained on 40GB of text (Radford et al., 2019). Due to the differences in the learning objective and data domain, a direct comparison between the pre-training data of GPT-2 and our datasets is difficult. In addition, the notion of tokens for graph learning models is not clearly defined. For example, for a graph with $n$ nodes and $e$ edges, one may regard the $n$ as the number of tokens. Further, since the edges represent the structural information in a graph, defining the number of tokens as $n + e$ might be more meaningful. This notion, however, might still not suffice to fully represent a meaningful token count, as we do not know beforehand the relative importance of node and edge tokens on pre-training graph learning models. Hence, we propose to additionally use the number of prediction targets or labels to compare the token count for two reasons: First, the number of labels has a direct influence on the number of gradient steps we can do per epoch, before the training data repeats. For example, pre-training on small graphs with many labels is likely favorable to pre-training on very large graphs with only very few labels. Second, when pre-training a transformer, such as GPT-2, on next-word prediction each input token can also be regarded as a prediction target, making the count of prediction targets an equally meaningful number for comparison.

Unfortunately, exact token counts for GPT-2 are not available. However, we estimate the number of pre-training tokens as follows. GPT-3, the successor model of GPT-2, was pre-trained on 570GB of text, resulting in 400B tokens (Brown et al., 2020b). Crucially, both GPT-2 and GPT-3 use Byte Pair Encodings (Sennrich et al., 2016), meaning that sentences are likely broken up into the same byte-pair encoded tokens. Hence, we estimate a GB-to-token conversion ratio of

$$\frac{400\text{B tokens}}{570\text{GB}} \approx \frac{0.7\text{B tokens}}{\text{GB}}$$

resulting in roughly 28B tokens for GPT-2.

For our proposed datasets, we present an overview over the available labels in Table 6 and arrive at a total of 13.04B across all mixes. Further, the number atoms and bonds per dataset can be found in Table 5. Here, we count a total of 4.5 atoms and bonds. While these numbers are still short of the 28B tokens of GPT-2, we argue that especially our labels can also be seen as more informative for the following reasons.

First, in contrast to the training data of GPT-2, where each label is a classification target, our labels comprise both classification and regression targets. In particular, regression targets might provide the model with much more fine-grained information, a factor which should be taken into account.

Second, the joint prediction of multiple labels, potentially even on multiple levels such as graph- and node-level, might provide additional information during training that is not easily captured with a simple count of the labels.

Third, there is evidence in the literature that supervised pre-training outperforms self-supervised methods when the domain of adaptation is similar to the training domain (Yang et al., 2020). We expect this to hold in our case since we have gathered a very large diversity of training tasks, increasing the likelihood that at least one training label is close to the desired fine-tuning label. Further, our dataset covers most of the molecules in PubChem, which means that they should cover a big part of the chemical space that is usually covered in drug-discovery projects.

## G    COMPUTATIONAL RESOURCES AND HARDWARE

**Hardware** Baseline results were trained on a Bow Pod-16 IPU system, made of 16 Bow IPU accelerators, with some additional results obtained using A100 and V100 GPUs.

**TOYMIX Computational resources** The results presented for the TOYMIX baselines used approximately 142 IPU hours of compute but we estimate that we used approximately 10 times that amount in the development stages for these models.

**LARGEMIX Computational resources.** The results presented for the LARGEMIX used approximately 600 IPU hours (for LARGEMIX, PCQM4M_N4, PCQM4M_G25) and 150 GPU hours (for PCBA_1328, L1000_VCAP, L1000_MCF7) of compute, with more compute used for development and hyperparameter search.

**ULTRALARGE Computational resources.** The results presented for the ULTRALARGE used approximately 500 GPU hours of compute, with more compute used for development and hyperparameter search.

## H    SCALING LAWS ON LARGEMIX

Table 7: Scaling Law Results for MPNN++ model on the proposed LARGEMIX datasets.

| model size (hidden size) | AVPR ↑ | | | $R^2$ ↑ | |
|---|---|---|---|---|---|
| | L1000_VCAP | L1000_MCF7 | PCBA_1328 | PCQM4M_G25 | PCQM4M_N4 |
| 1 million (64) | .4074 $\pm$ .0035 | .4173 $\pm$ .0036 | .2116 $\pm$ .0059 | .7737 $\pm$ .0014 | .7816 $\pm$ .0051 |
| 3 million (67) | .4060 $\pm$ .0004 | .4226 $\pm$ .0026 | .2462 $\pm$ .0054 | .7900 $\pm$ .0018 | .8189 $\pm$ .0066 |
| 10 million (128) | .4219 $\pm$ .0111 | .4475 $\pm$ .0138 | .2997 $\pm$ .0075 | .7974 $\pm$ .0004 | .8505 $\pm$ .0017 |
| 30 million (225) | .4608 $\pm$ .0185 | .4872 $\pm$ .0202 | .3405 $\pm$ .0122 | .7959 $\pm$ .0006 | .8614 $\pm$ .0014 |
| 100 million (416) | .4914 $\pm$ .0317 | .5235 $\pm$ .0369 | .3667 $\pm$ .0147 | .7981 $\pm$ .0004 | .8639 $\pm$ .0013 |
| 300 million (726) | .5280 $\pm$ .0004 | **.5657** $\pm$ .0024 | .3973 $\pm$ .0011 | **.8005** $\pm$ .0005 | .8658 $\pm$ .0008 |
| 1 billion (1331) | **.5285** $\pm$ .0005 | .5634 $\pm$ .0007 | **.4064** $\pm$ .0022 | .7995 $\pm$ .0007 | **.8680** $\pm$ .0007 |

**Experimental setting.** Here we aim to investigate the effect of scaling law (number of trainable parameters) on our proposed datasets with the MPNN++ model Masters et al. (2023). We tested the zero-shot scaling and up-scaling using the mup implementation Yang et al. (2021), meaning that no parameters were changed except the hidden dimensions being scaled. To our knowledge, this is the first time that mup was used in graph neural networks, so its behavior was not known. The Graphium library provides access to this valuable tool to the community, making investigation into scaling laws and previously unexplored model sizes for GNNs. To our delight, mup worked very well, showcasing another advantage of Graphium, and we can observe great scaling laws and model improvements up to a billion parameters.

**Results discussion.** As shown in Table 7, MPNN++ models with more parameters achieves significantly better results across the range of tasks. The top result is balded while the second place is placed with underline. Specifically, on the L1000_VCAP and L1000_MCF7, the avPR went up to $0.42 \rightarrow 0.53$ and $0.44 \rightarrow 0.56$ respectively, which is a large gain considering that 0.33 would be a random classifier. Further, PCBA_1328 went up from $0.30 \rightarrow 0.41$, again a large improvement considering the OGB-PCBA leaderboard (a smaller but similar dataset) is at 0.32. We further see results going down when the network is scaled down to 3M or 1M parameters.

**Hyperparameters.** The hyperparameters for the MPNN++ experiments are specified in Table 8 for reproducibility.

| Hyperparameter | Description |
|---|---|
| GNN depth | 16 |
| dropout | 0.01 |
| aggregation method | sum |
| MLP expansion ratio | 3 |
| node embedding combine method | concatenation |
| GNN normalization | batch normalization |
| any other normalization | layer normalization |
| gnn output dim | 512 |
| virtual node | log sum |
| gnn activation | gelu |
| any other activation | relu |
| Graph output NN graph depth, hidden | 2, 512 |
| Graph output NN node depth, hidden | 2, 256 |
| task heads depth, hidden | 2, 128 |
| batch size | 8192 |
| learning rate | lr = 0.008 with 5 epoch warm up |
| optimizer | Adam |
| epochs | 100 |

Table 8: Hyperparameters for the MPNN++ model in the scaling law experiment.

# I    ADDITIONAL VISUALIZATION ON MOLECULES

In this section, we provide additional visualizations of 24 molecules for L1000_MCF7 in Figure 7, PCBA_1328_1564K in Figure 8, PCQM4M_G25_N4 in Figure 9, QM9 in Figure 10, TOX21 in Figure 11 and ZINC12K in Figure 12.

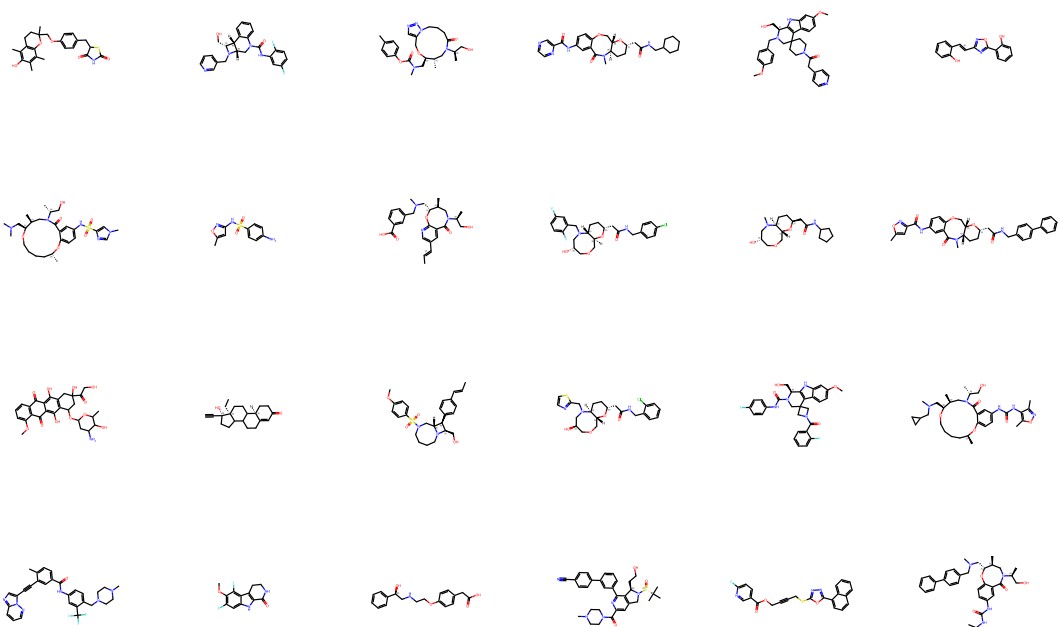

Figure 7: Molecule visualizations from L1000_VCAP dataset.

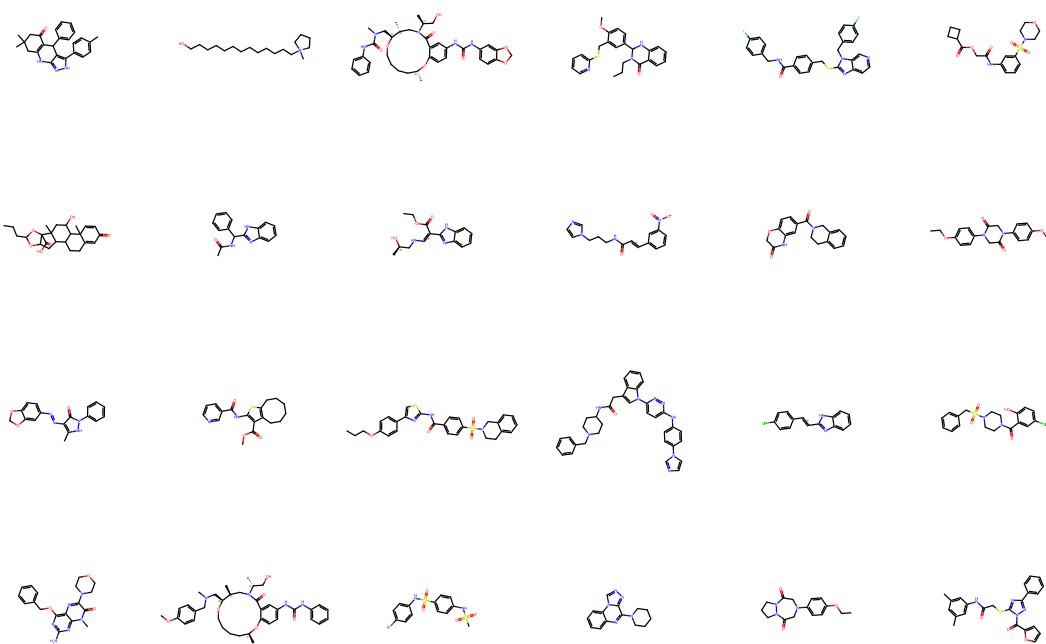

Figure 8: Molecule visualizations from PCBA_1328_1564K dataset.

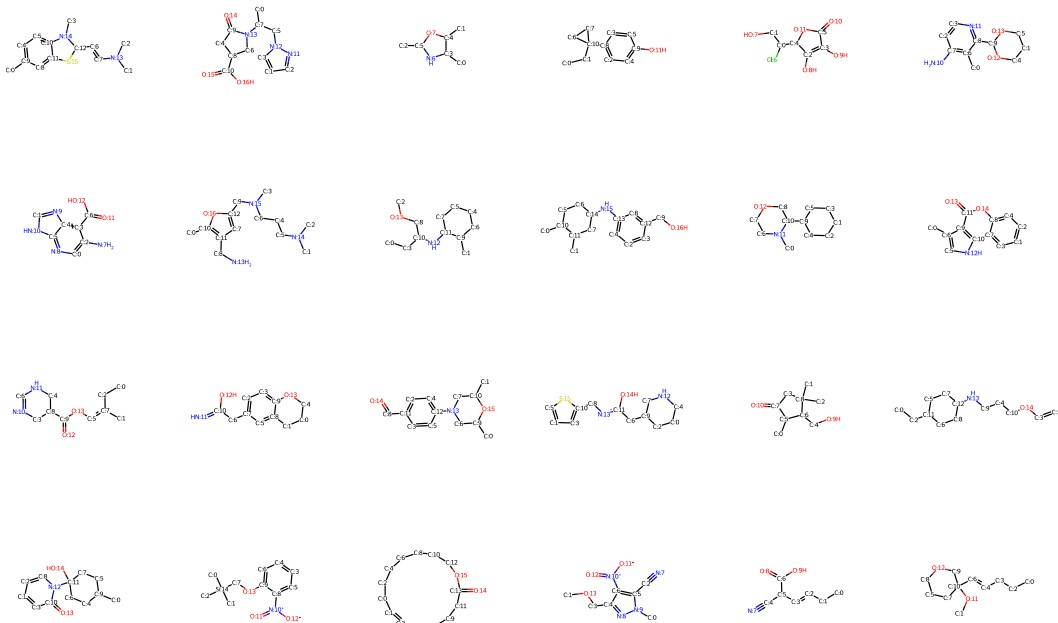

Figure 9: Molecule visualizations from PCQM4M_G25_N4 dataset.

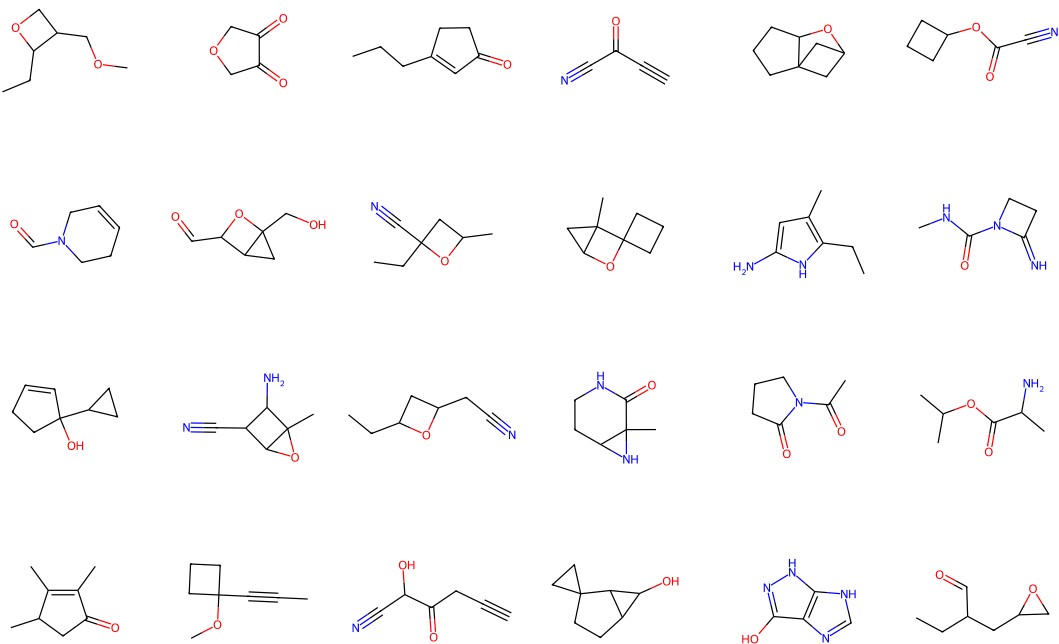

Figure 10: Molecule visualizations from QM9 dataset.

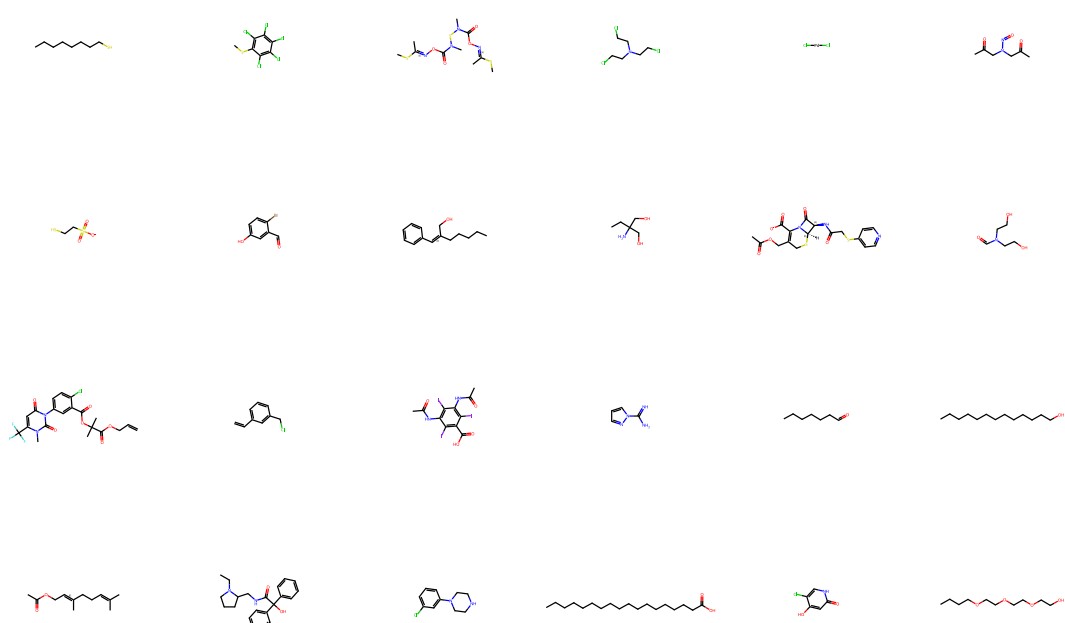

Figure 11: Molecule visualizations from TOX21 dataset.

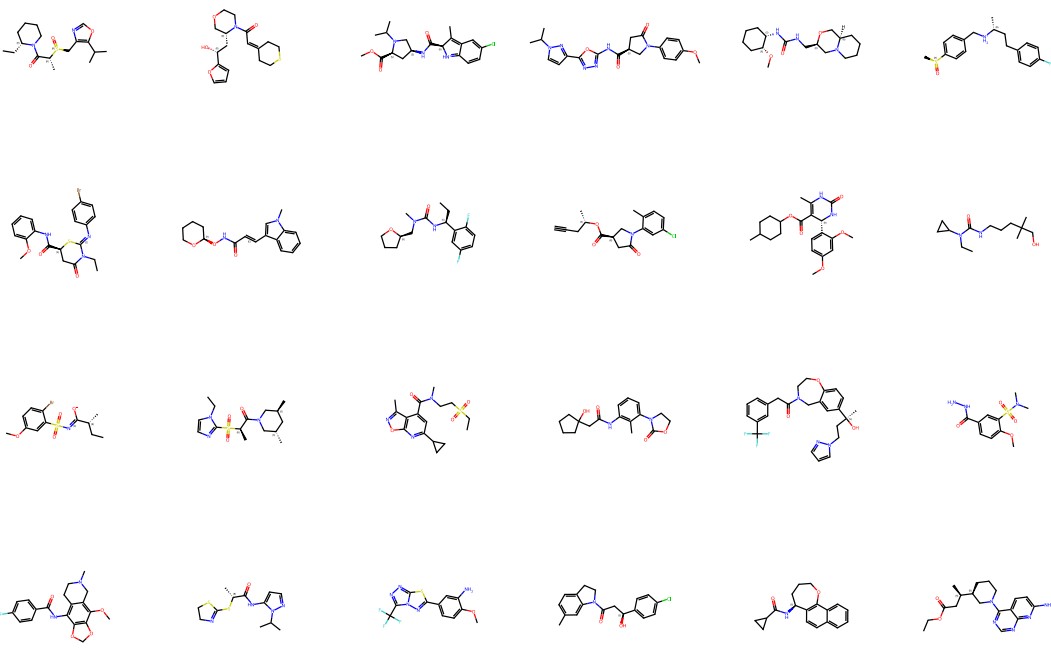

Figure 12: Molecule visualizations from ZINC12K dataset.

