# OpenReview forum: "Towards Foundational Models for Molecular Learning on Large-Scale Multi-Task Datasets"
_ICLR.cc/2024/Conference — ICLR 2024 poster_

### Official Review · Reviewer_43gK · 2023-10-30

**Soundness:** 3 good
**Presentation:** 3 good
**Contribution:** 3 good
**Rating:** 8
**Confidence:** 3

**Summary:**

This manusript proposed a curated dataset containing three subsets with different sizes. The whole datasets contains approximately 100 million molecules with more than 13 billion labels combining graph-level and node-level ones. Each molecules is accompanied with quantum mechancial properties calculated using either DFT or semi-emperical methods. A Python library called Graphium is built to facilitate the access and utilization of the proposed dataset for machine learning applications. Some tools such as handling missing data and label normalization in Graphium are designed to reduce the friction for users. Lastly, the authors provides benchmark results on the proposed dataset using common GNN models.

**Strengths:**

1. The proposed dataset is large and inclusive. In each one of the subset (ToyMix, LargeMix, and UltraLarge), at least one of the dataset contains 3D conformer of molecules and QM properties. As the 3D GNN models (e.g. Equivariant GNNs) are becoming increasingly important in the molecular machine learning community, 3D molecular conformer and corresponding labels are valuable.

2. The Graphium library can be handy for researchers who wants to join the molecular machine learning community but are halted by lack of experience with data processing and model training. With all datasets curated in this work included in the Graphium library, users can easily focus on innovating model architectures.

3. Many node-level labels are included in this dataset. Most conventional tasks in molecular machine learning are graph-level tasks such as molecular property prediction because of the scarcity of node-level labels.

**Weaknesses:**

1. One minor weakness is that the potential noise in the bioassay data. For the PCBA1328 dataset of LargeMix, the authors curated the data from PubChem by collecting molecules and their properties with regard to different bioassays. Due to the inherent noise associated with the data generation process with bioassay, model training (especially multitask training) can be difficult.

**Questions:**

1. For the PM6_83M, the authors mention that the QM properties of molecules are calculated using semi-empirical methods as reported in the original paper of PM6 paper. However, in secion D.4, the authors state that "Using the 3D conformation optimized by the DFT, we further computed the following 3D descriptors as labels". Can the authors confirm if they run DFT on top of the PM6 dataset or not? If they do run DFT, can they briefly discuss the DFT method they use here?

---

> ### Author Response · Authors · 2023-11-22
> **Thank you for the review**
>
> We are happy to see that the Reviewer found the datasets and library useful, and allow users to "easily focus on innovating model architectures". Below, we hope to answer the concerns and questions.
>
>
> ### W1. Noise in PCBA1328
> >One minor weakness is that the potential noise in the bioassay data. For the PCBA1328 dataset of LargeMix, the authors curated the data from PubChem by collecting molecules and their properties with regard to different bioassays. Due to the inherent noise associated with the data generation process with bioassay, model training (especially multitask training) can be difficult.
>
> **Answer**
>
> We thank the reviewer for pointing out this issue. Indeed, due to the nature of the data, there will be a base level of noise in the bioassay data. In this work, we curate the bioassays from existing data and utilize the reported flags from original authors. To denoise the data, the active and inactive labels from the original data are treated as 1 and 0 respectively while everything else is considered as NaN thus ignored rather than relying on the specific measured values which are more subject to experimental noise. Lastly, performing classification tasks on bioassay data helps reduce the noise by simplifying the labels to be binary though some effect of mislabeling might still be present.
>
>
> ### Q1. DFT or semi-empirical minimization
> >For the PM6_83M, the authors mention that the QM properties of molecules are calculated using semi-empirical methods as reported in the original paper of PM6 paper. However, in section D.4, the authors state that "Using the 3D conformation optimized by the DFT, we further computed the following 3D descriptors as labels". Can the authors confirm if they run DFT on top of the PM6 dataset or not? If they do run DFT, can they briefly discuss the DFT method they use here?
>
> **Answer**
>
> We thank you for reporting this error, we will correct it. The PCQM4M dataset has conformations minimized by DFT methods, with the labels also being computed using the same DFT. However, the PM6_83M dataset minimizes the conformations with semi-empirical PM6 method, and uses the same semi-empirical method to compute the labels.

---

> > ### Comment · Reviewer_43gK · 2023-11-22
> > **Response to revision**
> >
> > Thank you for your effort in clearing my concerns. My rating will remain as accept for this manuscript.

---

### Official Review · Reviewer_BQyU · 2023-10-31

**Soundness:** 3 good
**Presentation:** 2 fair
**Contribution:** 3 good
**Rating:** 6
**Confidence:** 4

**Summary:**

The authors have developed an extensive data repository and designed a graph learning library, “graphium” that supports the training of foundational models for molecular modeling. Authors have combined molecular datasets from various sources with labels for tasks targeting graph and node-level properties. The evaluation shows that training graph-based models on large amounts of quantum mechanical data improves the downstream performance of the model in resource-starved contexts, for example, modeling biological molecules.

**Strengths:**

+ The paper provides an extensive description of all the datasets that have been used and how they relate to each other in terms of what tasks they model and the labels provided. Particularly, the paper provides an in-depth explanation of (in the supplementary resources) how chemical and structural properties might relate to biological function, which can explain how properties learned in a quantum mechanical setting might help in a biological context.

+ Authors have made their work publicly available, including the datasets and the graph learning library, and providing multi-level, multi-label, and multi-task learning mechanisms. This form of modeling, as shown, is crucial in developing foundational models that generalize well to tasks with limited amounts of data available.

+ The authors discuss the positional encodings and distill the literature on relative and global positioning into their “graphium” data modeling pipeline.

**Weaknesses:**

+ The work is an important contribution to the ML and molecular learning communities. However, it restricts itself to data curation and performance results for different GNN-based models. The paper is missing crucial insights that might help inform the researchers interested in developing foundational models. For example, does the new dataset help improve other existing methods? Why the multi-task setting does not work well for all the datasets, and what are the data properties resulting in different performances for different datasets? How does this work translate into applications for the real-world setting?

+ The paper does not show a comparison against any other existing pre-trained molecular models. Moreover, it would also be helpful to highlight the performance against supervised SOTAs.

+ Although the paper mentions transformer-based models, such as Graphormer or GPS [1,2], in the supplementary section, it does not provide any results for them in the evaluation section. Showing results for them may be an essential consideration because they would show potentially better scaling to data than standard message passing, which tends to suffer from the problem of over-smoothing when the number of layers goes beyond three. Although the paper mentions this limitation and plans to do the comparison in the future, it's crucial to define how transformer-style models scale with the data that has been curated.

+ Continuing on the previous point, it would be nice to see a scaling curve that shows how increasing model complexity improves performance as we provide more data to it.


References:
[1] Benchmarking Graphormer on Large-Scale Molecular Modeling Datasets arXiv:2203.04810
[2] Recipe for a General, Powerful, Scalable Graph Transformer arXiv:2205.12454

**Questions:**

+ Authors mention that due to resource constraints, they only use a small fraction of the data to train the ULTRALARGE models; can they discuss how would the results change if the entire dataset is used? As mentioned by the authors, there is a skew towards the QM tasks.
Would the model overfit those tasks when trained with larger datasets and show degradation in performance for biological tasks?

+ It's unclear if the paper shows that the model can generalize to unseen tasks in the downstream setting.

+ Minor point: I am curious to see how the newer equivariant GNNs [3, 4] (specifically in the molecular modeling space) interact with having more data available to them.


References

[3] E(3)-Equivariant Graph Neural Networks for Data-Efficient and Accurate Interatomic Potentials https://arxiv.org/pdf/2101.03164.pdf
[4] Equivariant Graph Attention Networks for Molecular Property Prediction https://arxiv.org/pdf/2202.09891.pdf

---

> ### Author Response · Authors · 2023-11-22
> **Thank you for your thourough review**
>
> ### W1. Better understanding of how and why
> >The work ... restricts itself to data curation and performance results for different GNN-based models. The paper is missing crucial insights ... For example, does the new dataset help improve other existing methods? Why the multi-task setting does not work well for all the datasets, and what are the data properties resulting in different performances for different datasets? How does this work translate into applications for the real-world setting?
>
> **Answer**
>
> We thank you for these very important points. First, we want to reiterate that as part of the contributions of this work we do not just provide the curated datasets, but also Graphium, a library to enable large-scale training for molecules with various hardware. The Graphium library supports many SOTA models and positional encodings, mup for zero-shot scaling, and TDC for fine-tuning. The library being very modular, it allows to easily swap architectures, datasets, positional encodings, metrics, and benchmarks specifically to help study the very questions you are proposing.
>
> However, most of these questions are left for future work, as our paper proposes the first step to bring us there.
>
> **Why multi-task does not always work well**
>
> Regarding the question on why multi-task doesn’t work well on all datasets, we believe the main cause is underfitting, as a 10M parameter model is used to train on >3000 tasks jointly. This is supported by the reported results below, on single task, the test set is between 11% and 30% **worse** than train, but on multi-task, the test set is between 5% and 9% **better**. Again, looking at the scaling results, we can see that the performance on the PCBA_1328 goes up very significantly despite the multi-tasking.
>
> These points were clarified in the paper to better explain the disparity between the single-task and multi-task for PCBA_1328.
>
> |  | Train BCE Single task | Train BCE Multi-Task | Test BCE Single task | Test BCE Multi-Task |
> |--|--|--|--|--|
> | GCN | 0.0284 ± 0.0010 | 0.0382 ± 0.0005 | 0.0316 ± 0.0000 | 0.0349 ± 0.0002 |
> | GIN | 0.0249 ± 0.0017 | 0.0359 ± 0.0011 | 0.0324 ± 0.0000 | 0.0342 ± 0.0001 |
> | GINE | 0.0258 ± 0.0017 | 0.0361 ± 0.0008 | 0.0320 ± 0.0001 | 0.0341 ± 0.0001 |
>
>
> ### W2. Comparison to existing pre-trained models
> >The paper does not show a comparison against any other existing pre-trained molecular models. Moreover, it would also be helpful to highlight the performance against supervised SOTAs.
>
> **Answer**
>
> This is an important suggestion, but the aim of our current work is not to provide pre-trained models, but rather the datasets and library required to build foundational GNNs and benchmark them on TDC. Hence, our current baselines are not meant to be used for fine-tuning, and cannot be compared directly to self-supervised models.
>
>
> ### W3. Scaling Transformer models
> >Although the paper mentions transformer-based models, such as Graphormer or GPS [1,2], in the supplementary section, it does not provide any results for them in the evaluation section. ...
>
> **Answer**
>
> Referring to the general comment, we have shown some great scaling trends of up to 1B parameters with continuously improving results using MPNN++, the backbone behind the GPS++ winner of the OGB-LSC competition. We also aim to provide some baseline Transformers at the 10M parameter scale. However, in-depth comparison of the scaling laws of MPNN vs Transformer vs Hybrid is left for future work.
>
> Further, there are multiple literature showing that positional and structural encodings are more important than the GNN models itself, such as GPS[1], GPSE[2], LSPE[3], etc., and all models tested in the baseline use eigenvectors and random-walk encodings that are again behind the OGB-LSC competition win.
>
> [1] Rampášek, Ladislav, et al. "Recipe for a general, powerful, scalable graph transformer." Advances in Neural Information Processing Systems 35 (2022): 14501-14515.
>
> [2] Liu, Renming, et al. "Graph Positional and Structural Encoder." arXiv preprint arXiv:2307.07107 (2023).
>
> [3] Dwivedi, Vijay Prakash, et al. "Graph neural networks with learnable structural and positional representations." arXiv preprint arXiv:2110.07875 (2021).
>
> ### W4. Model scaling laws
> > ... it would be nice to see a scaling curve that shows how increasing model complexity improves performance as we provide more data to it.
>
> **Answer**
>
> We thank you for this suggestion, but we leave scaling laws analysis for the data for future work. In the general comment, we provided scaling laws for the model sizes.

---

> > ### Author Response · Authors · 2023-11-22
> > **Answering Additional questions**
> >
> > ###Q1. More illustrations
> >
> >
> > >... they only use a small fraction of the data to train the ULTRALARGE models; can they discuss how would the results change if the entire dataset is used? ... Would the model overfit those tasks when trained with larger datasets and show degradation in performance for biological tasks?
> >
> > **Answer**
> > This is a great point that we have carefully thought about in the design of Graphium. Indeed, regarding the question of imbalance and skew, Graphium currently supports the parameter `epoch_sampling_fraction` for each dataset. This parameter allows to sub-sample only a fraction of a dataset at any given epoch and can be used to re-balance the amount of molecules seen for each task. Therefore, we believe we can tune this parameter to avoid degradation on biological tasks.
> >
> > For example, selecting it to be `=0.05` would yield to about the same number of QM datapoints for both the UltraLarge and LargeMix datasets at each epoch, but would also yield to better generalization since it will take 20 epochs to see the entire dataset.
> >
> > In the interests of robustness and confidence of results provided the core experiments were priorities, and some experiments were not conducted due to finite compute resources. However, we agree this is an interesting future direction to explore.
> >
> > ### Q2. More illustrations
> >
> >
> > >It's unclear if the paper shows that the model can generalize to unseen tasks in the downstream setting.
> >
> > **Answer**
> >
> > Indeed, generalizing unseen tasks from a large pre-trained model is an important direction. In this work, we focus on curating the dataset and supporting library required to train a foundational model. We believe future work can pre-train with our proposed dataset and investigate the effectiveness on downstream unseen tasks.
> >
> > However, the Graphium library does include an easy access to the TDC API for fine-tuning with a tutorial in the file graphium/notebooks/finetuning-on-tdc-admet-benchmark.ipynb.
> >
> > ### Q3. More illustrations
> > > Minor point: I am curious to see how the newer equivariant GNNs [3, 4] (specifically in the molecular modeling space) interact with having more data available to them.
> > [3] E(3)-Equivariant Graph Neural Networks for Data-Efficient and Accurate Interatomic Potentials https://arxiv.org/pdf/2101.03164.pdf [4] Equivariant Graph Attention Networks for Molecular Property Prediction https://arxiv.org/pdf/2202.09891.pdf
> >
> > **Answer**
> > The Graphium library is meant for 2D GNNs, not 3D-GNNs or E-GNNs, so direct comparison cannot be made. Here, the QM dataset only comprises a single lowest-energy conformation, which is used to complement the information of the biological tasks. This is not suited for MD, which requires learning energies and forces outside of local-minima, and which requires the full 3D structure as input.

---

> > > ### Comment · Reviewer_BQyU · 2023-11-22
> > > **Thank you for the response**
> > >
> > > Thank you to the authors for the detailed responses to all my comments. While I still think that presenting some key insights from this work to the user community is important, I also agree that the contribution of this work is relevant. Therefore, I will raise my score to a "weak accept"

---

> > > > ### Author Response · Authors · 2023-11-23
> > > > **Thank you!**
> > > >
> > > > We would like to express our gratitude for your interest in our work, and for the important questions that you raised, enabling us to improve the quality of our work.

---

### Official Review · Reviewer_qFVJ · 2023-11-01

**Soundness:** 3 good
**Presentation:** 3 good
**Contribution:** 4 excellent
**Rating:** 8
**Confidence:** 2

**Summary:**

Having access to a free and open database of chemical compounds with their quantum features and biological activities. The main issue with the existing ones is the shortage of enormous data over various tasks, with many of them lacking experimental measures, or quantum and chemical properties. One know fact is that a little change in these properties can cause a major bioactivity behavior change. To end this, authors has collected a multiple number of datasets over molecules that will be a great help to build foundation models on this literature, e.g., models that are trained over multi-task and multi-level molecular datasets.

**Strengths:**

1. The contribution of this paper has been successfully delivered to the general audience. It will open doors to having more robust and representative models with the help of pre-training.
2. Advantages over non supervised methods for pre-training the model, such as contrastive learning, is important because of the "activity cliffs" phenomenon.
3. Many intricacies are considered into this, such as having both quantum and bioactivity features which are important for the future research.
4. Graphium library provides a reliable and tidy tool that gives a better pace to doing research.

**Weaknesses:**

1. It's not clear what are exact labels collected for each sample and how they can be useful.

**Questions:**

1. Please provides more samples of each dataset category for the sake of better illustration.

---

> ### Author Response · Authors · 2023-11-22
> **Thank you for the feedback!**
>
> We thank you for your feedback, for highlighting the importance of supervised learning for activity cliffs, of the combined quantum/bio tasks, and for finding the Graphium library "reliable and tidy".
>
> Below, we hope to answer your remaining concerns.
>
> ### W1. Labels description
> > It's not clear what are exact labels collected for each sample and how they can be useful.
>
> **Answer**
>
> Description of the labels are in `Appendix D DEEPER DIVE INTO THE LARGEMIX AND ULTRALARGE DATASETS`. For the PCBA_1328, it is impossible to provide a description of all 1328 labels, but the pubchem assay ID is still available in the column header of the CSV file so they are all traceable.
>
> If you have any specific questions on any specific dataset or label that Appendix D does not answer, please let us know.
>
> Q1. More illustrations
> >Please provides more samples of each dataset category for the sake of better illustration.
>
> **Answer**
>
> In Figure 2 of Appendix C, we provide 6 molecules per dataset. But to answer your request, we will re-do the figure to provide 24 molecules per dataset and span it across 2 pages.

---

### Official Review · Reviewer_jQ36 · 2023-11-04

**Soundness:** 3 good
**Presentation:** 3 good
**Contribution:** 3 good
**Rating:** 6
**Confidence:** 4

**Summary:**

The paper presents seven novel datasets that push the boundaries in both the scale and diversity of supervised labels for molecular learning. The authors also introduce the Graphium graph machine learning library to simplify the process of building and training molecular machine learning models. The paper argues that building effective foundational models for molecular modeling requires supervised training with both quantum mechanical (QM) descriptions and biological environment-dependent data. Overall, this paper aims to explore the possibilities of foundational models in molecular machine learning.

**Strengths:**

- The paper presents seven novel datasets that are orders of magnitude larger than the current state of the art, covering nearly 100 million molecules and over 3000 sparsely defined tasks, totaling more than 13 billion individual labels, currently the largest of their kind.

- The datasets are designed for the supervised training of foundation models by combining labels representing quantum and biological properties acquired through both simulation and wet lab experimentation. The diversity of labels facilitates efficient transfer learning and enables the construction of foundational models by improving their generalization ability for a wide range of downstream molecular modeling tasks.

- The paper introduces the Graphium graph machine learning library to facilitate efficient training on these extensive datasets. This library simplifies the process of building and training molecular machine learning models.

- The paper provides an anonymized repo for reproducibility commitment.

**Weaknesses:**

- The evaluated baselines are too simple/out-of-date in some sense. Since this paper is providing a data-driven platform for accelerating the research of molecular foundation models, it would be better to have included more SOTA baselines for comparison, which can help analyze the usefulness and effectiveness of this database better.

- It would be better to have a well-organized website with readable documents that can help understand the details and setups of different datasets/components better.

- More data splitting settings (e.g. mofit-based, system-based, scaffold-based, etc.) should be further studied. Now the OOD generalizability of GNNs across different molecule datasets still seem to be unclear.

**Questions:**

- I would be curious about how the models trained on larger datasets perform on smaller datasets (using the same task or different tasks). Also the performance of applying a well-trained model on a small dataset to larger datasets would be interesting to study to understand the few-shot learning ability of those models over these datasets.

---

> ### Author Response · Authors · 2023-11-22
> **Thank you for the review**
>
> We would like to thank Reviewer jQ36 for their thorough review, and for highlighting the strengths related to dataset sizs and library building. We hope to answer their questions and concerns below.
>
> ### W1. More SOTA baselines
> >The evaluated baselines are too simple/out-of-date in some sense. Since this paper is providing a data-driven platform for accelerating the research of molecular foundation models, it would be better to have included more SOTA baselines for comparison, which can help analyze the usefulness and effectiveness of this database better.
>
> **Answer**
>
> We would like to point the reviewer to the general comment. We have added a baseline for MPNN++ (the backbone of the GPS++ and winner of the OBG-LSC competition), and aim to provide results for graph Transformer. Further, there are multiple literature showing that positional and structural encodings are more important than the GNN models itself, such as GPS[1], GPSE[2], LSPE[3], etc., and all models tested in the baseline use eigenvectors and random-walk encodings that are again behind the OGB-LSC competition win.
>
> [1] Rampášek, Ladislav, et al. "Recipe for a general, powerful, scalable graph transformer." Advances in Neural Information Processing Systems 35 (2022): 14501-14515.
>
> [2] Liu, Renming, et al. "Graph Positional and Structural Encoder." arXiv preprint arXiv:2307.07107 (2023).
>
> [3] Dwivedi, Vijay Prakash, et al. "Graph neural networks with learnable structural and positional representations." arXiv preprint arXiv:2110.07875 (2021).
>
>
> ### W2. Well-organized website
> >It would be better to have a well-organized website with readable documents that can help understand the details and setups of different datasets/components better.
>
> **Answer**
> We thank the reviewer for this proposition, and would like to mention that we do have a website! The website contains different tabs (Overview, Baseline, API, Tutorials, Design, Datasets, Pretrained Models, CLI, Contribute, License).
>
> For anonymity reasons, we cannot share the website but invite you to build it locally:
>
> - Install mkdocs package `pip install mkdocs`
> - Go to the main graphium directory
> - Run it on a local server `mkdocs serve`
> - CTRL+Click on the link given after the compilation
>
>
> ### W3. More data splitting settings
> >More data splitting settings (e.g. mofit-based, system-based, scaffold-based, etc.) should be further studied. Now the OOD generalizability of GNNs across different molecule datasets still seem to be unclear.
>
> **Answer**
>
> We thank the reviewer for this proposition, but would like to reiterate that the objective of the dataset is to pre-train a large model, not to test the OOD. However, we do support direct finetuning on TDC to test the generalizability of the model, but this is left for future work. A tutorial on how to finetune on TDC is available in the file graphium/notebooks/finetuning-on-tdc-admet-benchmark.ipynb.
>
> ### Q1. Scaling data and few-shot learning
> >I would be curious about how the models trained on larger datasets perform on smaller datasets (using the same task or different tasks). Also the performance of applying a well-trained model on a small dataset to larger datasets would be interesting to study to understand the few-shot learning ability of those models over these datasets.
>
> **Answer**
>
> We agree with the reviewer here, but leave this question for future work.
>
> At the moment, we provide the datasets, the library with support for zero-shot scaling thanks to mup, show some very interesting scaling trends in the general comment, and provide support for downstream fine-tuning on TDC.

---

> > ### Comment · Reviewer_jQ36 · 2023-11-22
> > **Thanks**
> >
> > Thank you for addressing my concerns. I'll keep my rating as it is.

---

### Official Review · Reviewer_Zdj5 · 2023-11-05

**Soundness:** 3 good
**Presentation:** 3 good
**Contribution:** 2 fair
**Rating:** 3
**Confidence:** 4

**Summary:**

This paper present seven novel datasets with different sizes, to facilitate the application and development of molecular foundation model. Multiple baselines are conducted and some experimental results and analysis are provided on several downstream tasks. Furthermore, Graphium graph machine learning library is proposed to offer more convenient implementations of multi-task and multi-level machine learning methods.

**Strengths:**

1. This paper compiles a comprehensive benchmark for the evaluation of molecular models. The dataset covers a large variety of molecules and labels at different levels, with particularly comprehensive and complete data for quantum-related tasks.
2. The Graphium library appears promising as a convenient tool for researchers or developers to perform fine-tuning on various tasks using different backbones.

**Weaknesses:**

1. The diversity of this benchmark remains a problem. For example, most of the contribution comes from the quantum related property data, which cover a limited part in molecular property prediction.
2. The paper just collected existing data instead of creating new data, which limits the significance.
3. Baselines and discussions are not thorough, leading to unconvincing results.
4. Some empirical settings are not clear or strict, making the comparison or benchmark not solid.

**Questions:**

1. Although the molecular numbers reach the million scale, it appears that the majority of the contribution comes from quantum data. The tasks lack diversity, and there is an imbalance among the three scales of data (toymix, largemix, ultralarge). For instance, the largest ultralarge dataset solely comprises quantum tasks. It would be more comprehensive if it included more biological tasks, such as ADMET properties.
2. Seems that this paper did not create any new data, instead, they only collected some existing data to organize to different levels or merge different datasets, right? In this case, the novelty and contribution is limited.
3. From Table 1, we observe that training on multitasks aids in improving the performance of ZINC12k and Tox21, but it doesn't yield the same benefit for Quantum tasks (QM9). What could be the reason behind this phenomenon?
4. If adding the Quantum tasks enhances the performance of biological tasks, why does the multitask performance for the PCBA_1328 dataset in the largemix dataset at Table 2 not surpass that of the single task setting? The data size may not be the final answer and further analysis may be necessary.
5. The baselines are too simplistic, as they only include three different types of GCNs. More models involving various types of self-supervised methods, which are essential for a foundation model, should be included in the experiments as baselines for discussion.
6. If this dataset is aimed at the foundation model, it could potentially achieve the best performance on various tasks(both quantum and bio-related tasks) across different scales. However, this paper only tests various networks on data of different scales dataset, and it lacks related experiments or discussions.
7. More sophisticated split functions beyond random, such as scaffold, can be explored, as the out-of-distribution (OOD) problem is common in large models and requires careful study.

---

> ### Author Response · Authors · 2023-11-22
> **We hope to address the weaknesses**
>
> We would like to clarify that the goal of this paper is to construct the largest public collection of molecular data, as well as an optimized library consisting of SOTA GNNs, for building foundational models for molecular learning (as stated in the abstract).
>
> From reviewer’s Zdj5 response, we believe that they perceived our work as a paper focusing on benchmarking different GNN architecture for molecular learning, which is not the case, hence there is a disagreement between the review and the work’s intention, which we hope to clarify better in our response.
> ### W1. Benchmark Diversity
> >The diversity of this benchmark remains a problem. For example, most of the contribution comes from the quantum related property data, which cover a limited part in molecular property prediction.
>
> **Answer**
>
> We respectfully disagree with the reviewer. We believe our **curated** datasets are highly diverse in the non-quantum tasks as well. For instance, the PCBA_1328 dataset combines 1328 assays covering a vast task space and the L1000 data combines 998 gene perturbations across 2 cell lines. Hence, there’s a total of 3324 bio-assays in the LargeMix dataset, which we curated ourselves into ML ready format fitting for GNNs.  To the best of our knowledge, these alone are more diverse than any public 2D molecular learning datasets in the literature.
>
> It is also worth noting that due to the time intensive nature of results derived from experimental, rather than computational sources, such datasets are significantly smaller than their computational counterparts. While the reviewer is absolutely correct that if more experimental / biological data were available it would be preferable, we argue that the combination of large quantum datasets augmented with a smaller portion of high quality experimental data allows a model to learn suitably general embeddings, and believe the results presented show this.
>
> ### W2. Only Collects Existing Data
> >The paper just collected existing data instead of creating new data, which limits the significance.
>
> **Answer**
>
> First, we note that the datasets are only one of the contributions, as we also propose an open-source *Graphium* library that contains the SOTA graph network models for molecules and positional/structural encodings.
>
> Second, in the field of molecular learning, collecting novel data, especially for non-quantum tasks is extremely expensive and often involves costly real-world experiments. In addition, available data is often not in a ML ready format and requires extensive data processing with domain experts to prepare them for training in graph neural networks. In this work, we curated 3 datasets for the LargeMix and 1 dataset for the Ultra-large, with labeled features as a starting point for building foundational models on molecules. The datasets cover nearly 100 million molecules and over 3000 sparsely defined tasks totaling more than 13 billion individual labels. There is significant value in the curation of these data in simple tabular formats.
>
> ### W3. Not Enough Baselines
> >Baselines and discussions are not thorough, leading to unconvincing results.
>
> **Answer**
>
> The focus of this work is to construct the largest public collection of molecular data and the Graphium library which supports graph ML on the proposed dataset. Our objective is not for building a benchmark to compare many GNN architectures.
> The baselines provided are intended to demonstrate both the soundness of the datasets and multi-task learning in a machine learning context, and provide a reference point for other researchers developing new models for expected performance of simple and reproducible models.
>
> Although more thorough experiments are left for future work both internally and to the community, we aim to add results for the Graph Transformers. We further added some scaling law results using the MPNN++ architecture, the backbone of the GPS++ model that won the OGB-LSC competition, and we observed an impressive scaling trend up to 1B parameters.
>
> ### W4. Setting not clear
> >Some empirical settings are not clear or strict, making the comparison or benchmark not solid.
>
> **Answer**
>
> We provide details on our experimental settings for TOYMIX, LARGEMIX and ULTRALARGE in detail in Section 4.1, 4.2 and 4.3 respectively. Appendix E also provides details on the graphium library for multi-level, multi-task and multi-label learning, the modeling and more. We would be glad to further clarify any specific concerns for the reviewer.

---

> > ### Comment · Reviewer_Zdj5 · 2023-11-22
> >
> > Thank you for your response.
> > To be clear, my question about unclear empirical setting is that, we are not sure whether for each task, all the baselines employ the same setting, e.g. data split strategy or scaffold spit strategy, which are usually crucial factors to produce different results.

---

> > > ### Author Response · Authors · 2023-11-22
> > > **Clarifying the splits**
> > >
> > > >my question about unclear empirical setting is that, we are not sure whether for each task, all the baselines employ the same setting, e.g. data split strategy or scaffold spit strategy
> > >
> > > Thank you for following up on your question. Each of the datasets within LargeMix and ToyMix has its own train/val/test split, with the indices defined in a pickle file. When training in a multi-task setting, each dataset is split into train/val/test according to its own indices; then all the train are merged together, all the val together, and all the test together. This means that the single-task setting and multi-task setting have seen exactly the same labels for a given task. In LargeMix, we also ensured that the molecules in the test set on one task were never seen in the training set of another task.
> > >
> > > We reiterate that all splits are random since the goal is to cover as much chemical space for pre-training, not to test OoD generalization on the current set of tasks. This is a typical strategy employed for pre-training foundational models.

---

> ### Author Response · Authors · 2023-11-22
> **Thank you for the questions**
>
> ### Q1. Lack of biological task
> > ... It would be more comprehensive if it included more biological tasks, such as ADMET properties.
>
> **Answer**
>
> We want to point out that our contributed datasets contain a total over 3000 tasks thus are highly diverse in nature. When examining the balance between biological tasks and quantum tasks, one must first consider the data availability. Note that ADMET properties are generally only found in very small amounts as it requires lab experiments. The well-known TDC benchmark[1] contains a number of ADMET properties however each property only has 500-6000 molecules and are notorious for their curation issues, see [“Getting Real with Molecular Property Prediction](https://tinyurl.com/mr2mvavt) and [“We Need Better Benchmarks for ML in DD”](https://tinyurl.com/cdhht757).
>
> Therefore, we omitted the ADMET at the moment due to the issues with dataset curation and sizes. We also reiterate that we are proposing datasets for pre-training at a large scale, not a task-specific benchmark.
>
> [1] Huang, Kexin, et al. "Therapeutics Data Commons: Machine Learning Datasets and Tasks for Drug Discovery and Development." 2021.
>
> ### Q2. did not create any new data
>
> > Seems that this paper did not create any new data, instead, they only collected some existing data...
>
> **Answer**
>
> We refer the reviewer to the detailed response we provided for W2.
>
> ### Q3. explanation
> >... multitasks aids in improving the performance of ZINC12k and Tox21, but it doesn't yield the same benefit for Quantum tasks (QM9)...
>
> **Answer**
>
> We thank the reviewer for raising this point. First, note that performance benefits on the ZINC12k and Tox21 are **more interesting** than QM9. This is because, in practice, biological tasks have significantly less training data than quantum tasks, and the goal of building a foundational model with a 2D GNN is to improve the performance of low-data biological tasks through the joint pre-training with quantum tasks which we observe in this case.
>
> Second, the models use the same amount of parameters to train on both multitask and the sole QM9 task thus the parameters are shared across other tasks for the multitask model. Third, QM9 originally has a much higher volume of data, so it benefits less from the other tasks.
>
> ### Q4. explanation
> >... performance for the PCBA_1328 dataset in the largemix dataset at Table 2 not surpass that of the single task setting? ...
>
> **Answer**
>
> We thank the reviewer for this observation and will clarify it in the paper. We believe the main cause is underfitting, as a 10M parameter model is used to train on >3000 tasks jointly. Thus, we argue that longer training (capped at 100 epochs) and more parameters (capped at 10M) will benefit the multi-task model more.
>
> This hypothesis is supported by the reported results below, on single task, the test set is between 11% and 30% **worse** than train, but on multi-task, the test set is between 5% and 9% **better**. Again, looking at the scaling results, we can see that the performance on the PCBA_1328 goes up very significantly despite the multi-tasking.
>
> |  | Train BCE Single task | Train BCE Multi-Task | Test BCE Single task | Test BCE Multi-Task |
> |--|--|--|--|--|
> | GCN  | .0284 ± 0.0010  | .0382 ± 0.0005 | .0316 ± 0.0000 | .0349 ± 0.0002 |
> | GIN  | .0249 ± 0.0017  | .0359 ± 0.0011 | .0324 ± 0.0000 | .0342 ± 0.0001 |
> | GINE | .0258 ± 0.0017 | .0361 ± 0.0008 | .0320 ± 0.0001 | .0341 ± 0.0001 |
>
> ### Q5. More Baseline
> >The baselines ... only include three different types of GCNs...
>
> **Answer**
> We would like to refer the reviewer to the general comment, where we added baselines for MPNN++, the backbone model behind the OGB-LSC challenge winner, and scaling laws.
> And additionally clarify for the reviewer that the aim is to provide baselines for the datasets, not to optimize performance on this new dataset collection.  .
>
>
> ### Q6. Performance of foundation model
> >If this dataset is aimed at the foundation model, it could potentially achieve the best performance on various tasks...
>
> **Answer**
>
> We refer the reviewer to the general comment, where we provided scaling laws showcasing significant improvements up to 1B parameters.
>
>
> ### Q7. more splits
> >More sophisticated split functions beyond random, such as scaffold, can be explored...
>
> **Answer**
>
> We would like to reiterate that we are not proposing a benchmark paper, but rather datasets to pre-train a model, and pre-training is known to work better with random split as it allows the model to better see the desired chemical space. However, the Graphium library does include an easy access to the TDC API where OOD generalization can be tested using the various splits proposed. A tutorial on how to finetune on TDC is available in the file graphium/notebooks/finetuning-on-tdc-admet-benchmark.ipynb.

---

### Author Response · Authors · 2023-11-22
**Thank you! And more results**

We are glad to see that the reviewers recognize that our work provides a comprehensive benchmark for the evaluation of molecular models (reviewer Zdj5), orders of magnitude larger than the current state of the art (reviewer jQ36) and will be a great help to build foundation models (reviewer qFVJ). Further, reviewers Zdj5, jQ36, qFVJ, 43gK all agree that our Graphium library is a promising tool for training ML models on molecules.

This comment aims to answer common questions regarding better baselines and studying scaling laws.

**Better baselines**

We continued to improve the baseline results with model sizes around 10M parameters. We found that the MPNN++ (the backbone of the GPS++ , winner of the OGB-LSC competition) performed best, and further optimized it by searching over 2000 configurations. With the MPNN++ the following results are obtained, which are very significant improvements on the previously reported baselines (with an R2 of ~0.25 on the node-level PCQM4M_n4, and an AvPR of 0.25 on the PCBA_1328).

| model_size (hidden_size) | l1000_vcap (avpr) | l1000_mcf7 (avpr) | pcqm4m_g25 (r2) | pcqm4m_n4 (r2) | pcba_1328 (avpr) |
|--------------------------|-------------------|-------------------|-----------------|----------------|------------------|
| 10M  (128)               | .4219 ± .0111     | .4475 ± .0138     | .7974 ± .0004   | .8505 ± .0017  | .2997 ± .0075    |

The selected parameters are:

```
gnn_depth: 16
dropout: 0.01
aggregation_method: [sum]
mlp_expansion_ratio: 3
node_combine_method: concat
gnn_normalization: batch_norm
any_other_normalization: layer_norm
gnn_out_dim: 512
virtual_node: logsum
gnn_activation: gelu
any_other_activation: relu
Graph_output_nn.graph.depth / hidden =2 / 512
Graph_output_nn.node.depth / hidden =2 / 256
task_heads.*.depth / hidden =2 / 128
batch_size=8192
lr=0.008 with 5 epoch warmup
optimizer: Adam
epochs=100
```

**Scaling laws on LargeMix**

First, we reiterate that the focus of the current work is not about studying scaling laws, but in the hopes of answering the reviewer’s concerns, we have studied the zero-shot scaling of our model. To the best of our knowledge, this is the first evidence supervised training of molecular GNNs scaling to 1B parameters.

We tested the zero-shot scaling and up-scaling using the [mup](https://github.com/microsoft/mup) implementation, meaning that no parameters were changed except the hidden dimensions being scaled. To our knowledge, this is the first time that mup was used in graph neural networks, so its behavior was not known.
The Graphium library provides access to this valuable tool to the community, making investigation into scaling laws and previously unexplored model sizes for GNNs.
To our delight, mup worked very well, showcasing another advantage of Graphium, and we can observe great scaling laws and model improvements up to a billion parameters.

As can be seen in the results detailed below, which will be added to the paper, this more expressive model achieves significantly better results across the range of tasks.
Specifically, on the L1000_VCAP and L1000_MCF7, the avPR went up to 0.42->0.53 and 0.44->0.56 respectively, which is a large gain considering that 0.33 would be a random classifier. Further, PCBA_1328 went up from 0.30->0.41, again a large improvement considering the OGB-PCBA leaderboard (a smaller but similar dataset) is at 0.32. We further see results going down when the network is scaled down to 3M or 1M parameters.

| model_size (hidden_size) | l1000_vcap (avpr) | l1000_mcf7 (avpr) | pcqm4m_g25 (r2) | pcqm4m_n4 (r2) | pcba_1328 (avpr) |
|--------------------------|-------------------|-------------------|-----------------|----------------|------------------|
| 1M   (36)                | .4074 ± .0035      | .4173 ± .0036      | .7737 ± .0014   | .7816 ± .0051  | .2116 ± .0059    |
| 3M   (67)                | .4060 ± .0004      | .4226 ± .0026      | .7900 ± .0018   | .8189 ± .0066  | .2462 ± .0054    |
| 10M  (128)               | .4219 ± .0111      | .4475 ± .0138      | .7974 ± .0004   | .8505 ± .0017  | .2997 ± .0075    |
| 30M  (225)               | .4608 ± .0185      | .4872 ± .0202      | .7959 ± .0006   | .8614 ± .0014  | .3405 ± .0122    |
| 100M (416)               | .4914 ± .0317      | .5235 ± .0369      | .7981 ± .0004   | .8639 ± .0013  | .3667 ± .0147    |
| 300M (726)               | .5280 ± .0004      | .5657 ± .0024      | .8005 ± .0005   | .8658 ± .0008  | .3973 ± .0011    |
| 1B   (1331)              | .5285 ± .0005      | .5634 ± .0007      | .7995 ± .0007   | .8680 ± .0007  | .4064 ± .0022    |


**Graph Transformer results**

Based on the significant improvements from the MPNN++ model, we are working on training the GPS++ model, which combines the MPNN++ with an attention layer. GPS++ is the winner of the OGB-LSC challenge, thus considered the state of the art for 2D GNNs. We aim to add the results to the paper in the near future.

---

> ### Author Response · Authors · 2023-11-22
> **Manuscript Updated**
>
> Based on the constructive feedbacks from the reviewers as well as recent experiments, we have revised the manuscript to reflect the following changes (changes marked by blue):
>
> * Added Scaling law experiment details in Appendix H including experimental setting, results discussion and hyperparameters for reproducibility.
> * Added details and screenshot of the documentation website for Graphium in Appendix E.2.
> * Additional visualization of 24 molecules from each dataset added in Appendix I.
> * Updated the [anonymized repo](https://anonymous.4open.science/r/graphium-5F12/README.md) with documentations in `docs/` and notebook example for fine tuning on TDC ADMET example.
> * Clarifications added based on feedback from Reviewer Zdj5 Q4 and Reviewer 43gK Q1.

---

### Meta-Review · Area_Chair_GsVD · 2023-12-07

**Metareview:**

This work introduces seven new datasets for molecular machine learning, covering nearly 100 million molecules and over 3000 tasks, and presents the Graphium library for building and training models, with initial results indicating potential benefits from multi-task and multi-level training. Reviewers generally appreciate introducing of such a dataset to the community.

**Justification For Why Not Higher Score:**

The paper focuses more on the introduction of the new datasets, does not include a comparison of existing pre-training models.

**Justification For Why Not Lower Score:**

A large and diverse dataset is indeed necessary for building a foundation model on Molecular Learning.

---

### Decision · Program_Chairs · 2024-01-16

Accept (poster)